# Anaerobic fungi in the tortoise alimentary tract illuminate early stages of host-fungal symbiosis and *Neocallimastigomycota* evolution

Carrie J. Pratt [1], Casey H. Meili [1], Adrienne L. Jones[1], Darian K. Jackson[1], Emma E. England[1], Yan Wang [2], Steve Hartson[3], Janet Rogers[3], Mostafa S. Elshahed [1] & Noha H. Youssef [1] ✉

Anaerobic gut fungi (AGF, *Neocallimastigomycota*) reside in the alimentary tract of herbivores. While their presence in mammals is well documented, evidence for their occurrence in non-mammalian hosts is currently sparse. Culture-independent surveys of AGF in tortoises identified a unique community, with three novel deep-branching genera representing >90% of sequences in most samples. Representatives of all genera were successfully isolated under strict anaerobic conditions. Transcriptomics-enabled phylogenomic and molecular dating analyses indicated an ancient, deep-branching position in the AGF tree for these genera, with an evolutionary divergence time estimate of 104-112 million years ago (Mya). Such estimates push the establishment of animal-*Neocallimastigomycota* symbiosis from the late to the early Cretaceous. Further, tortoise-associated isolates (T-AGF) exhibited limited capacity for plant polysaccharides metabolism and lacked genes encoding several carbohydrate-active enzyme (CAZyme) families. Finally, we demonstrate that the observed curtailed degradation capacities and reduced CAZyme repertoire is driven by the paucity of horizontal gene transfer (HGT) in T-AGF genomes, compared to their mammalian counterparts. This reduced capacity was reflected in an altered cellulosomal production capacity in T-AGF. Our findings provide insights into the phylogenetic diversity, ecological distribution, evolutionary history, evolution of fungal-host nutritional symbiosis, and dynamics of genes acquisition in *Neocallimastigomycota*.

Microbial communities play a crucial role in the digestive process in herbivores by mediating the breakdown of substrates recalcitrant to their hosts' enzymes[1–3]. The establishment of herbivore-microbiome nutritional symbiosis was associated with the evolution of dedicated digestive chambers, e.g., enlarged hindgut, diverticula, and rumen, and longer feed retention times to improve the efficiency of the digestion process[4–7]. A complex community of microorganisms in the herbivorous gastrointestinal tract (GIT) breaks down plant biomass

[1]Department of Microbiology and Molecular Genetics, Oklahoma State University, Stillwater, OK, USA. [2]Department of Biological Sciences, University of Toronto Scarborough, Toronto, ON, Canada. [3]Department of Biochemistry and Molecular Biology, Oklahoma State University, Stillwater, OK, USA. ✉e-mail: noha@okstate.edu

into absorbable end products[3]. So far, greater emphasis has been placed on the study of bacterial and archaeal members of the community[8–16], compared to microbial eukaryotes (protozoa and fungi). Nevertheless, the role of eukaryotes in the herbivorous gut is increasingly being recognized[17–21].

The anaerobic gut fungi (AGF, *Neocallimastigomycota*) are integral and ubiquitous constituents of the GIT community in mammalian herbivores[22–27]. Notably, while chiefly investigated in mammalian hosts, microbiome-enabled herbivory, and associated GIT structural features conducive to AGF establishment also occur in multiple non-mammalian herbivores. One of the potential non-mammalian AGF hosts are tortoises, members of the family *Testudinidae*, order *Testudines*[28]. Tortoises are terrestrial herbivores that feed on grains, leaves, and fruits, possess an enlarged cecum[29], retain food for extremely long time frames (12–14 days)[29], and rely on hindgut fermentation[30,31].

Here, we challenge the prevailing mammalian-centric narrative of AGF distribution and evolutionary history by the identification, isolation, and characterization of ancient, deep-branching AGF taxa from the tortoise GIT. The discovery of these tortoise-associated AGF (T-AGF) demonstrates that AGF evolution as a distinct fungal phylum predates the rise of mammalian herbivory post the K-Pg extinction event; previously regarded as the defining event driving *Neocallimastigomycota* evolution and establishment of herbivores-fungal nutritional symbiosis[27,32]. Finally, we assess trait evolution patterns in these deep-branching taxa in comparison to other mammalian-associated AGF and demonstrate that massive horizontal gene transfer (HGT) events driving CAZyome expansion in the *Neocallimastigomycota* have occurred mostly in mammalian-, but not T-AGF lineages.

## Results

### Anaerobic gut fungal diversity and community structure in tortoises

Culture-independent analysis identified the occurrence of AGF in all tortoise samples examined ($n = 11$, Table S1, Dataset S1). A distinct community composition pattern was observed, with sequences affiliated with three AGF genera (NY54, NY56, and NY36) being highly prevalent, representing (either individually or collectively) >90% of sequences encountered in 9/11 samples (Dataset S1, Fig. 1A). Candidate genus NY54 was the most ubiquitous being identified in all tortoise samples, as well as the most abundant making up >90% of the AGF community in 6 samples (Pancake, Impressed, Egyptian, Indian star, one Galapagos, and one Sulcata tortoise) and >50% of the AGF community in one sample (Burmese star tortoise). Candidate genus NY56 was encountered only in 3/11 samples, and in only one of these (the Texas tortoise), it constituted >90% of the AGF community while making up only a minor fraction (<1%) in the other two samples. Similarly, candidate genus NY36 was less ubiquitous, being only encountered in two samples, and constituting >90% of the community in one sample. In contrast to their collective abundance in tortoise fecal samples, two out of the three tortoise-associated genera (NY36 and NY56) were seldom identified in reference mammalian fecal samples, while the third genus (NY54) exhibited a higher level of occurrence (Fig. 1B). Regardless of their observed pattern of ubiquity, the three tortoise-associated genera constituted a minor component of the community in mammalian samples, whenever encountered (Fig. 1B).

Phylogenetic analysis using the D2 domain of the LSU rRNA placed NY54, NY56, and NY36 as three distinct, deeply branching lineages within the *Neocallimastigomycota* tree (Fig. 1C), with the closest relative being the genus *Khoyollomyces* (Fig. 1C). Sequences for each of the three genera were identified across various tortoises (Fig. S1), suggesting ready cross-host colonization. The observed low intra-genus sequence divergence estimates for all three genera suggest

a low level of speciation (Fig. 1D). Quantitative PCR was conducted to estimate the abundance of AGF in tortoise fecal samples. AGF load (expressed as ribosomal copy number per gram feces) was invariably low in all tortoise fecal samples examined. Loads were much higher in AGF canonical mammalian hosts, e.g., cattle, sheep, goats, and horses (Fig. 1E).

Assessment of alpha diversity patterns indicated that tortoise samples, on average, harbored a significantly less diverse AGF community when compared to placental mammals ($p < 0.04$) (Fig. 2A). Lower AGF alpha diversity was observed in 8 tortoise fecal samples, while alpha diversity in three samples (including the two samples exhibiting a community dominated by AGF genera typically encountered in mammals) was as high or higher than in mammals (Fig. 2A). Community structure assessment using PCoA (constructed using weighted Unifrac) confirmed the clear distinction between tortoise and mammalian AGF mycobiomes (Fig. 2B, C). DPCoA ordination plots (Fig. 2D) showed that the abundance of the tortoise-associated genera NY36, NY54, and NY56, and the paucity of all other AGF was responsible for the observed pattern of community structure distinction.

However, it is important to note that all the tortoise samples studied here originated from zoo settings. To normalize for any effect the domestication status might have on diversity estimates, we compared the AGF community in the 11 tortoise samples to 11 mammalian samples (Dataset S2B) that were obtained from the same zoo. Again, tortoise samples harbored a significantly less diverse AGF community when compared to mammalian zoo samples ($p < 0.04$) (Fig. S2a), with a unique community structure (Fig. S2b) that could be significantly explained by the host class (adonis $p = 0.001$, $R^2 = 0.49$).

### Isolation of tortoise-associated AGF genera

Isolation efforts from tortoise fecal samples yielded twenty-nine different isolates (Table S2, Fig. S3). Amplification and sequencing of the D1/D2 region of the LSU rRNA confirmed that the obtained isolates are identical to sequences encountered in culture-independent surveys. Representative isolates belonging to the candidate genera NY54 and NY36 have been successfully maintained and characterized as *Testudinimyces* gen. nov. and *Astrotestudinimyces* gen. nov., respectively[33]. *Testudinimyces* and *Astrotestudinimyces* isolates were enriched at, and displayed optimum growth temperatures, of 30 °C and 39 °C, respectively. On the other hand, despite repeated successful enrichment and isolation rounds (conducted at 39 °C), isolates belonging to candidate genus NY56 have been hard to maintain as viable cultures for subsequent analysis. The names *Testudinimyces* and *Astrotestudinimyes* will henceforth be used to describe cultured strains belonging to NY54 and NY36 in the manuscript. Enrichments at 42 °C did not yield any visible fungal growth.

### Phylogenomic and molecular clock timing analysis

Transcriptomics-enabled phylogenomic analysis placed isolates belonging to the genera *Testudinimyces* and *Astrotestudinimyces* as two distinct, early branching basal lineages in the *Neocallimastigomycota* tree (Fig. 3, Fig. S4). Molecular clock timing suggests that T-AGF evolved in the early Cretaceous, with the earliest divergence time estimated at 112.19 Mya (genus *Astrotestudinimyces* lineage), with the 95% Highest Probability Density (HPD) interval at 95.94-129.98 Mya. A divergence estimate of 104.43 Mya was observed for the genus *Testudinimyces* lineage (95% HPD: 89.37–120.82 Mya). Such estimates push the *Neocallimastigomycota* evolution by ~45 Mya, since prior efforts timed the phylum evolution at 67 Mya[27,32], and indicate that *Neocallimastigomycota* evolution predates the K-Pg extinction event and subsequent evolution of mammalian herbivorous families (e.g., *Equidae, Bovidae, Cervidae*) as well as grasses (*Poaceae*), previously regarded as the defining process forging AGF evolution into a distinct fungal phylum[27,32].

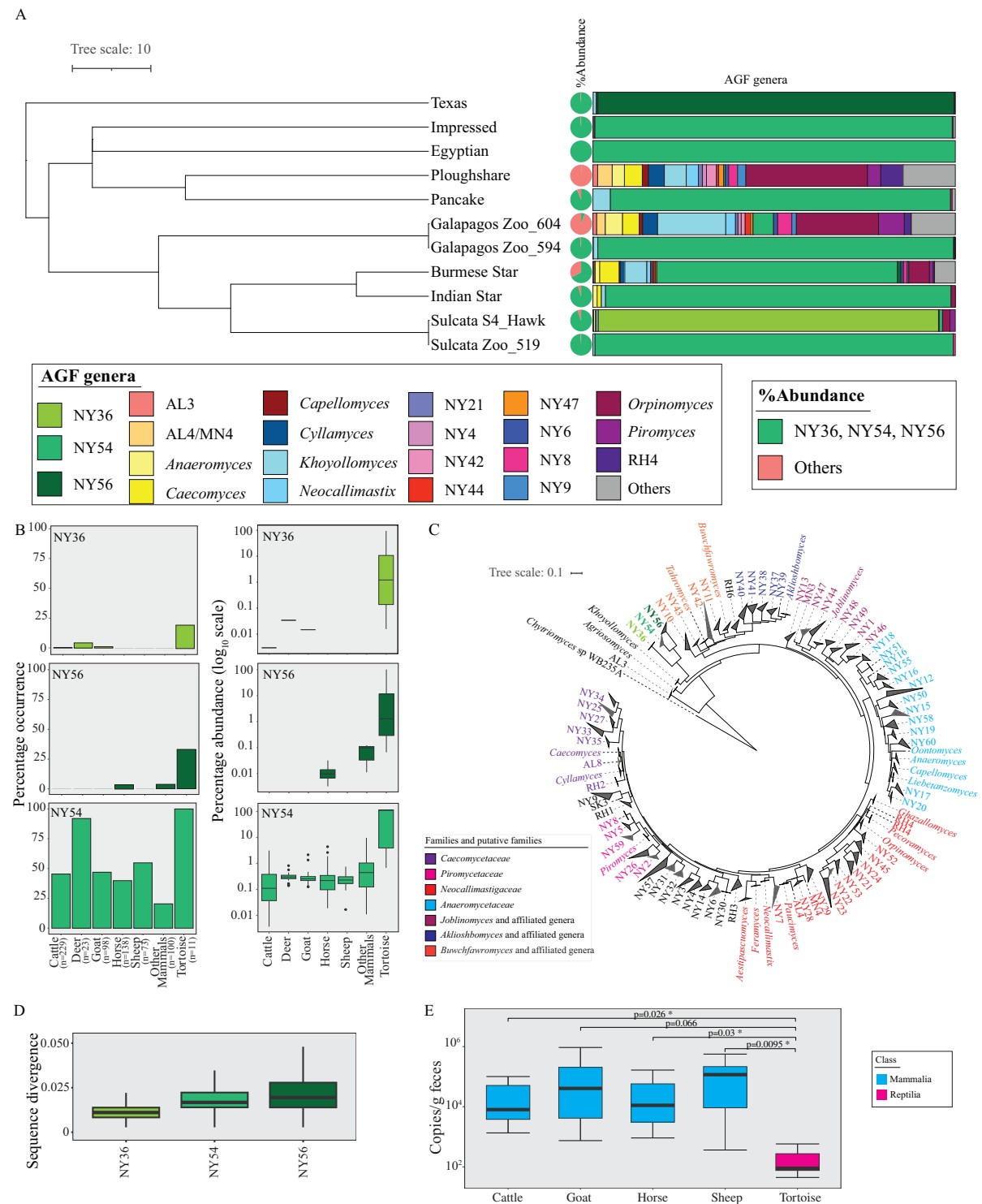

As previously demonstrated, AGF taxa previously isolated from mammals (M-AGF) possess reduced mitochondria in the form of a double-membrane hydrogenosome, an organelle whose main functions are substrate-level phosphorylation and $H_2$ production[34–36]. Comparison of the predicted hydrogenosomal content of one representative of each T-AGF genus (strains T130A.3 and B1.1), to an M-AGF strain (*Orpinomyces joyonii* strain AB3) showed similar patterns between M-AGF and T-AGF, with a near complete hydrogenosomal protein import system, chaperones/co-chaperones, mitochondrial peptidases, and mitochondrial transporters (Dataset S3). Hydrogenosomes in M-AGF are the sites of multiple metabolic processes,

including pyruvate metabolism, ATP production via substrate-level phosphorylation, regeneration of reduced electron carriers, some amino acid biosynthesis, fatty acid biosynthesis, and Fe-S cluster assembly. Our comparative analysis showed a similar pattern in T-AGF predicted hydrogenosome, with all these functions described above predicted to be hydrogenosomal. Interestingly, M-AGF hydrogenosomes lack a genome and all matrix proteins are nuclear-encoded, cytosol-biosynthesized, then imported into the hydrogenosome. AGF's closest relative aerobic fungi have genome-harboring mitochondria (mitogenome) that can independently replicate, transcribe, and translate mitochondrial DNA[37]. Fungal mitogenomes typically

**Fig. 1 | AGF diversity and community structure in Tortoises. A** Community composition in the 11 tortoise fecal samples studied. The tortoise phylogenetic tree was downloaded from timetree.org. The pie chart to the right shows the total percentage abundance of the three tortoise-affiliated genera (NY54, NY36, and NY56) (green) versus other AGF genera (peach). AGF community composition for each tortoise sample is shown to the right as colored bars corresponding to the legend key. **B** Percentage occurrence (left) and percentage abundance (right) of the three tortoise-affiliated genera in previously studied cattle, white-tail deer, goats, horses, sheep, and other mammals[27], as well as in the 11 tortoise samples studied. The number of individuals belonging to each animal species is shown on the X-axis. Color code follows the key in (**A**). **C** Maximum likelihood phylogenetic tree constructed from the alignment of the D1/D2 region of the LSU rRNA genes and highlighting the position of the three tortoise-affiliated genera in relation to all previously reported cultured and uncultured AGF genera. Genera are color-coded by family or putative family, and the three tortoise-affiliated genera are shown in green boldface. **D** Distribution of sequence divergence within each genus. Data are derived from 129284, 7525594, and 586689 distances for NY36, NY54, and NY56 respectively. **E** AGF load (determined using qPCR and expressed as copy number/g fecal sample) in the 11 tortoise samples studied here in comparison to ten individual cattle, goats, sheep, and horses selected. Significance is shown above the boxplots and corresponds to the two-sided Student *t*-test *p* value. Source data for Fig. 1B–D are provided as a Source Data file. Boxplots in (**B**, **D**, **E**) extend from the first to the third quartile and the median is shown as a thick line in the middle. The whiskers extending on both ends represent variability outside the quartiles and are calculated as follows: Minimum whisker = minimum quartile − 1.5 × inter-quartile range; Maximum whisker = maximum quartile + 1.5 × inter-quartile range. All points outside the box and whiskers are outliers.

encode RNAs or proteins involved in translation (e.g., a set of tRNAs and some small and large ribosomal subunit RNAs), as well as components of the ETS and oxidative phosphorylation, among other components[38]. Surprisingly, the predicted hydrogenosomal content of T-AGF (but not M-AGF) included several peptides assigned to DNA repair and recombination, several translation factors and proteins, and several mRNA and tRNA biogenesis proteins hinting at a possible presence of an organelle genome (Dataset S3). Further analysis is needed to confirm this possibility (for example identification of mitochondrial rRNA and tRNA genes, or phylogenetic congruency of mitochondrial T-AGF genes with their aerobic counterparts). However, the nature of the current dataset (transcriptomic rather than genomic) and the fact that ribodepletion was applied prior to poly-A transcripts selection for mRNA enrichment prior to RNA-seq would prohibit such analysis currently. Future availability of genomes from T-AGF isolates should allow further analysis of this interesting possibility.

## A curtailed carbohydrate-active enzyme machinery in tortoise-associated AGF

Preliminary comparative transcriptomic analysis (Supplementary text, Fig. S5) indicated that T-AGF lack many gene clusters (*n* = 1699) encountered in all currently available mammalian-isolated AGF (M-AGF) isolates. Interestingly, a significant proportion (55.13%) of these gene clusters encoded metabolic functions, with a high proportion of carbohydrate metabolism (49.63% of metabolic functions gene clusters) and, more specifically, an enrichment of Carbohydrate-Active enZymes (CAZymes) (46% of carbohydrate metabolism clusters) (Fig. S6). Further, representatives of the genera *Testudinimyces* (strain T130A.3) and *Astrotestudinimyces* (strain B1.1) demonstrated a slower ability to grow on (microcrystalline) cellulose, failed to grow on xylan, and exhibited a relatively more limited capacity for carbohydrates metabolism compared to reference M-AGF (Fig. S7 and in ref. 33). This pattern strongly suggests a curtailed machinery for plant biomass degradation in tortoise-associated, compared to M-AGF isolates.

Comparative analysis demonstrated that T-AGF harbor a significantly reduced CAZyome compared to M-AGF (Student *t*-test *p* = 0.0011), with only 0.5 ± 0.11% of the predicted peptides assigned to GH, CE, and PL families, compared to 1.3 ± 0.61% in mammalian isolates transcriptomes (Dataset S4). Specifically, T-AGF transcriptomes harbored a significantly lower number of distinct transcripts assigned to the families primarily associated with cellulose and hemicellulose metabolism, e.g., cellulase GH families GH5, GH9, the xylanase families GH10, GH11, GH16, GH45, the cellobiohydrolase families GH6, GH48, the β-glucosidase family GH3, the β-xylosidase family GH43, the α-amylase family GH13, the acetyl xylan esterases families CE1, CE2, CE4, and CE6 (Wilcoxon adjusted *p* < 0.03) (Fig. 4). The same pattern was obtained when T-AGF transcriptomic datasets (*n* = 7) were compared to four subsets of the M-AGF transcriptomic datasets with a similar number (*n* = 6–9) (Fig. S8, Table S3).

## Limited horizontal gene transfer in tortoise-associated AGF

Interestingly, many of the CAZyme families lacking or severely curtailed in T-AGF have previously been shown to be acquired by AGF via horizontal gene transfer (HGT) (Fig. S6)[39]. To determine whether this reflects a broader pattern of sparse HGT occurrence in the entirety of T-AGF genomes, we quantified HGT occurrence and frequency in T-AGF transcriptomes. Our analysis (Table 1) identified a total of only 35 distinct HGT events (with an average of 0.16 ± 0.05% of transcripts in the sequenced T-AGF transcriptomes). This value is markedly lower than the 277 distinct HGT events previously reported from M-AGF transcriptomes[39]. Interestingly, within the limited number of HGT events identified in T-AGF, the majority (30/35) were also identified in M-AGF[39]; and virtually all of which (29/30) had the same HGT donor (Table 1). This consistency in observed HGT events (i.e., same function, same donors) between both T-AGF genera as well as between T-AGF and M-AGF taxa argues against potential bacterial or archaeal contamination as a source of such transcripts.

Prior work has suggested the prevalence of metabolic functions in genes acquired by HGT in M-AGF[39]. This pattern held true for T-AGF, with (29/35) of the identified HGT events encoding a metabolic function. Several HGT-acquired metabolic genes in T-AGF were involved in processes enabling anaerobiosis. Specifically, these genes mediated functions such as recycling reduced electron carriers via fermentation (aldehyde/alcohol dehydrogenases and d-lactate dehydrogenase for ethanol and lactate production from pyruvate), de novo synthesis of NAD via the bacterial pathway (L-aspartate oxidase NadB), the acquisition of the oxygen-sensitive ribonucleoside triphosphate reductase class III (anaerobic ribonucleoside triphosphate reductase nrdD) and of squalene-hopene cyclase, catalyzing the cyclization of squalene into hopene during biosynthesis of tetrahymanol (that replaced the molecular $O_2$-requiring ergosterol in the cell membranes of AGF) (Table 1). Few additional HGT-acquired metabolic genes encoded CAZymes (Table 1). However, the number of HGT-acquired CAZyme genes in T-AGF was minor (13 events representing an average of 10.81 ± 4.17% of the total CAZyome in sequenced transcriptomes) compared to the massive acquisition of CAZymes by HGT previously reported in M-AGF (a total of 72 events representing 24.62–40.41% of the overall CAZyome)[39].

## Cellulosomal production capacity in tortoise versus mammalian AGF

Anaerobic fungi produce cellulosomes, extracellular structures that function as multienzyme complexes that synergistically break down plant biomass into fermentable sugars[40,41]. Many AGF-produced CAZymes localize to the cellulosomes. A non-catalytic dockerin domain (NCDD) similar to carbohydrate-binding module family 10 (CBM10) is usually associated with cellulosome-bound genes in anaerobic fungi and typically docks the enzymes to cohesin domains housed in a large scaffolding protein (scaffoldin), that in-turn anchors the entire structure to the cell wall[26]. We hypothesized that the

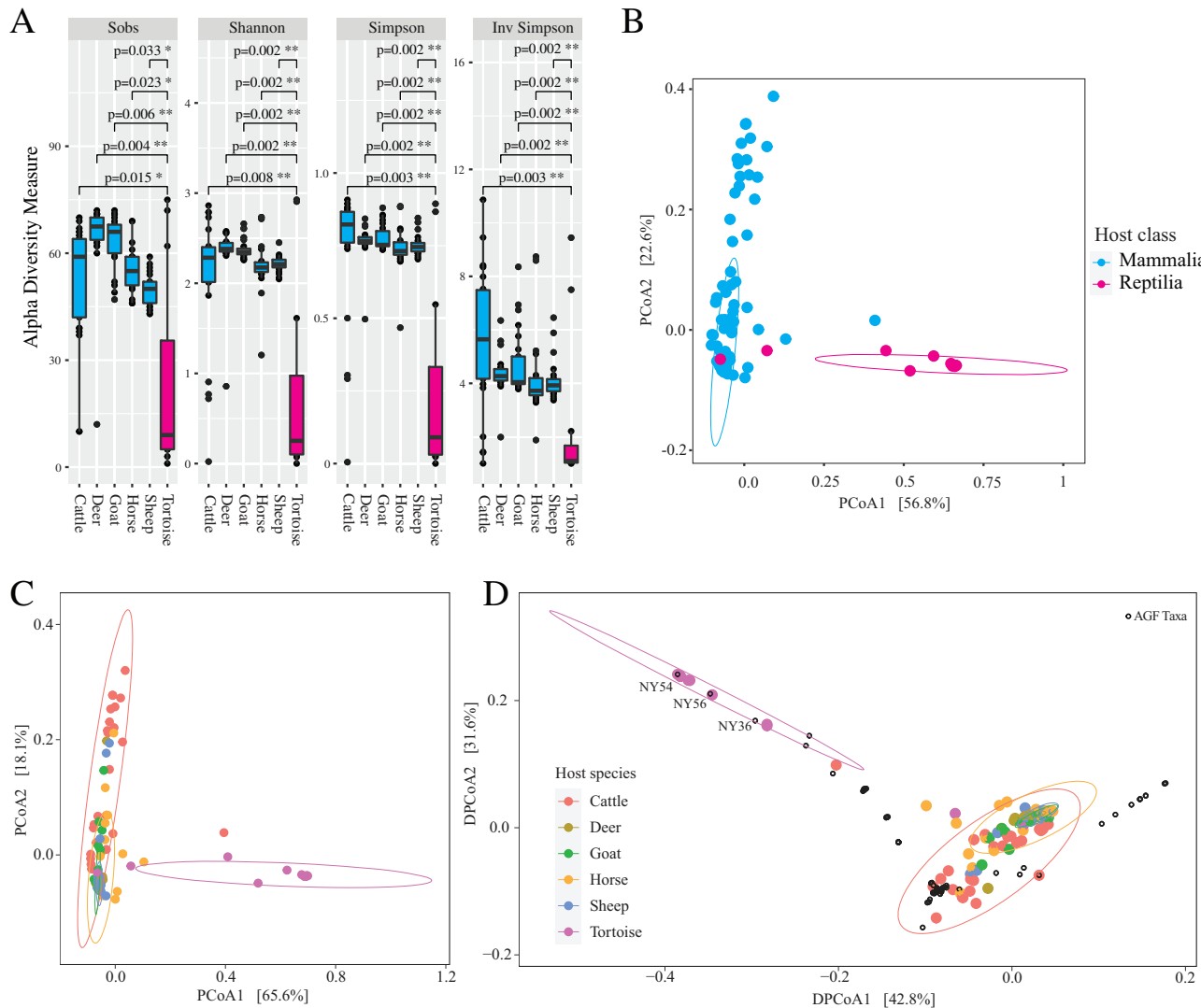

**Fig. 2 | Patterns of AGF alpha and beta diversity in the 11 tortoise samples studied in comparison to a subset of mammalian hosts previously studied[27] (Dataset S2A). A** Box and whisker plots showing the distribution of 4 alpha diversity measures (observed number of genera (Sobs), Shannon, Simpson, and Inverse Simpson) for the different animal species. Results of two-sided Wilcoxon signed rank test for pairwise comparison of tortoise (pink) alpha diversity indices to mammals (cyan; cattle ($n = 25$), deer ($n = 24$), goats ($n = 25$), horses ($n = 25$), and sheep ($n = 25$)) are shown above the boxplots. * Denotes p values of $0.01 < p < 0.05$ and ** denotes $0.001 < p < 0$. Boxplots in Fig. 1B, D, E extend from the first to the third quartile and the median is shown as a thick line in the middle. The whiskers extending on both ends represent variability outside the quartiles and are calculated as follows: Minimum whisker = minimum quartile − 1.5 × inter-quartile range; Maximum whisker = maximum quartile + 1.5 × inter-quartile range. All points outside the box and whiskers are outliers. **B**, **C** Principal coordinate analysis (PCoA) plot based on the phylogenetic similarity-based index weighted Unifrac. The percentage variance explained by the first two axes is displayed on the axes, and ellipses encompassing 95% of variance are displayed. Samples and ellipses are color-coded by host class (**B**), and host species (**C**). Some of the circles representing tortoise samples might not be apparent due to overlap with other data points. **D** Double principal coordinate analysis (DPCoA) biplot based on the phylogenetic similarity-based index weighted Unifrac. The percentage variance explained by the first two axes is displayed on the axes, and ellipses encompassing 95% of variance are displayed. Samples and ellipses are color-coded by host species. AGF genera are shown as black empty circles, and the three tortoise-affiliated genera are labeled. Source data are provided as a Source Data file.

observed differences in gene content (Fig. S6), CAZyme repertoire (Fig. 4, Fig. S8, Table S3, Dataset S4), secretome content (Supplementary text, Fig. S9), and HGT frequency (Table 1) between T-AGF and M-AGF would result in a differential cellulosomal production capacity (assessed as all peptides predicted to be extracellular and harbor an NCDD, as previously suggested[42–44]). Within the transcriptomes of representatives of T-AGF genera *Testudinimyces* (strain T130A) and *Astrotestudinimyces* (strain B1.1), predicted peptides with high sequence similarity (>27.34% aa identity), and close phylogenetic affiliation (Fig. 5A) to *Neocallimastigomycota* scaffoldin protein ScaA were identified (5 copies in T130A, and 38 copies in B1.1 transcriptomes equivalent to 0.03, and 0.14% of total transcripts). As well, a total of 91

and 183 transcriptome-predicted peptides possessing a NCDD and predicted to be extracellular were identified in T130A and B1.1, respectively (equivalent to 1.16 and 0.34% of total transcripts). NCDD-harboring predicted peptides encoded CAZymes ($n = 43$, and 72, respectively), spore coat protein CotH ($n = 6$, and 9, respectively), carbohydrate-binding modules ($n = 34$, and 87, respectively), expansins ($n = 1$, and 3, respectively), and other functions including hydrolases, and phosphatases (Fig. 5B). For comparative purposes, we sequenced and analyzed the transcriptome of a reference M-AGF isolate (*Orpinomyces joyonii* strain AB3). Similar to previously reported M-AGF, e.g., *Pecoramyces*[42], *Caecomyces*[45], *Piromyces*, *Neocallimastix*, and *Anaeromyces*[44], strain AB3 harbored a larger number of scaffoldin-

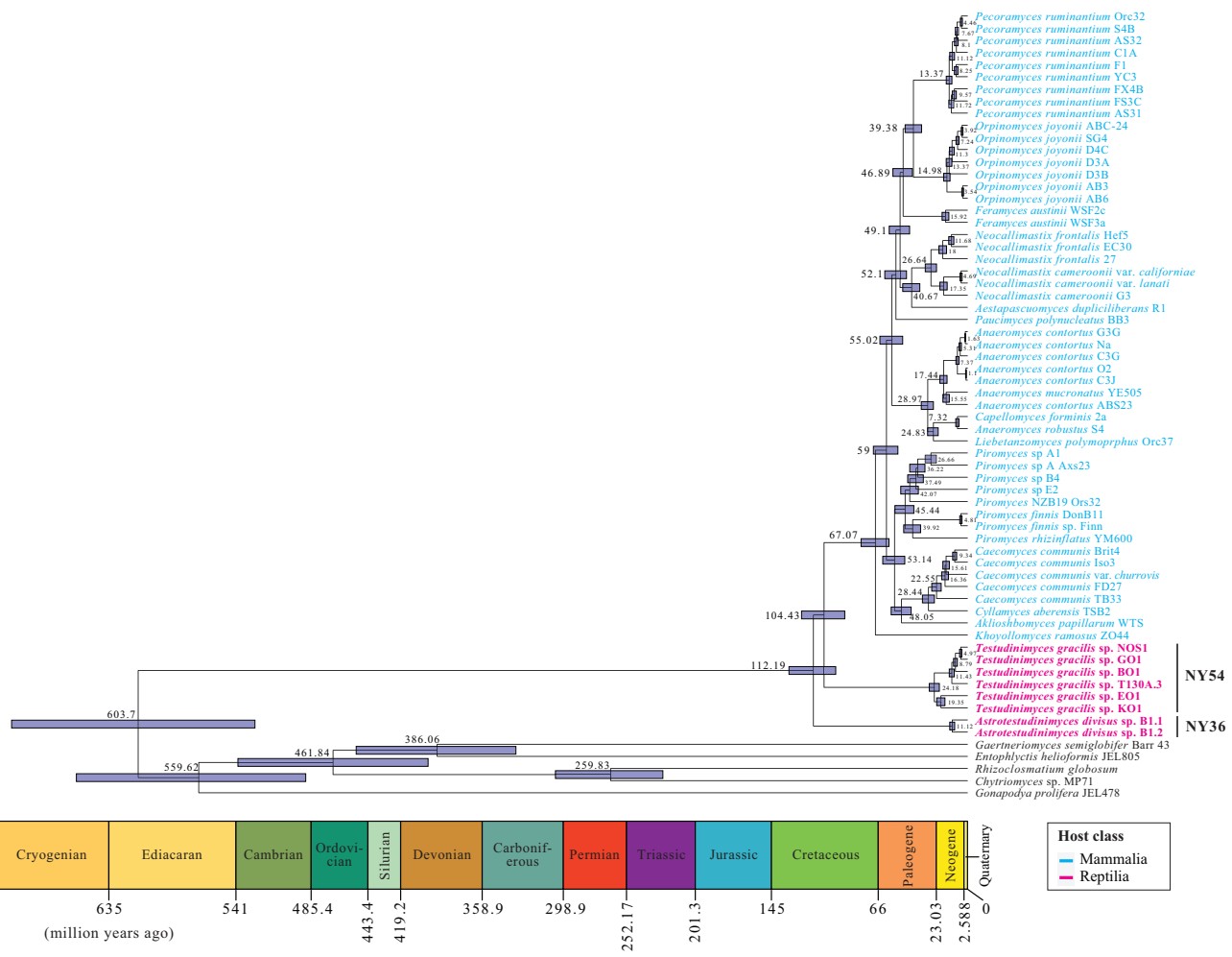

**Fig. 3 | Bayesian phylogenomic maximum clade credibility (MCC) tree of *Neocallimastigomycota* with estimated divergence time.** The isolate names are color-coded by host class as shown in the legend. Strains belonging to the two T-AGF genera are shown in boldface, and the taxa label is shown to the right. All clades above the rank of the genus are fully supported by Bayesian posterior probabilities. The 95% highest probability density (HPD) ranges (blue bars) are denoted on the nodes, and the average divergence times are shown. Geological timescale is shown below.

predicted peptides ($n = 24$, equivalent to 0.146% of total transcripts), and extracellular predicted peptides possessing a NCDD ($n = 316$, equivalent to 1.92% of total transcripts). Extracellular NCDD-harboring predicted peptides in strain AB3 encoded CAZymes ($n = 134$), spore coat protein CotH ($n = 30$), carbohydrate-binding modules ($n = 118$), and expansins ($n = 3$) (Fig. 5B). Further, in addition to the overall lower number of NCDD-harboring peptides in T-AGF compared to M-AGF, clear differences were also observed in the relative composition of the CAZyme component of their predicted cellulosomes. In general, a minor representation of CEs and GH10 components in T-AGF cellulosomes, when compared with the M-AGF representative strain AB3, was observed (Fig. 5B). On the other hand, an exclusive representation of GH45 in strain B1.1, and high and exclusive representation of PL1 in strain T130A, in comparison to strain AB3 cellulosome, was observed (Fig. 5C).

To confirm the translation, secretion, and cellulose-binding affinity of predicted cellulosomal proteins (scaffoldins and NCDD-containing peptides), shotgun proteomics was conducted on the total, and cellulose-bound fractions of representatives of *Astrotestudinimyces* (strain B1.1), and *Testudinimyces* (strain T130A) (Dataset S5). Of the 221 and 96 proteins predicted to be cellulosomal-bound in transcriptomics analysis of strains B1.1 and T130A, respectively, 172 and 57 proteins were identified in the proteomics dataset, confirming their translation (Dataset S6, Fig. S10). Of these, 169 and 50 proteins were identified in the cellulose-bound fraction. Further, all or the majority of these proteins were identified in higher intensity in the cellulose-bound fraction (169 and 46 proteins, respectively), with intensity ratios (intensity in cellulose-bound fraction: intensity in biomass fraction) exceeding 5 in 94% and 90% of the proteins (Dataset S6, Fig. S10).

## Discussion

Assessment of the AGF community in tortoises demonstrated that three genera (*Testudinimyces*, *Astrotestudinimyces*, and NY56) represent the majority of AGF in most samples examined (Fig. 1A). However, the remaining two tortoise samples had an AGF community that is highly similar to AGF communities typically observed in mammalian samples (Figs. 1A, 2B–D)[27]. The fact that few tortoise samples were dominated by a community similar to that observed in mammals, as well as the fact that representatives of the three T-AGF genera were identified in mammalian samples (Ref. 27, Fig. 1B, albeit with lower frequency and relative abundance) argues for the occurrence of cross-colonization of AGF between tortoise and mammals. The implications of this cross-colonization and host jumping ability is significant, since it would enable various fungal taxa to survive through extinction of various herbivorous hosts.

We argue that the observed AGF community composition patterns (Fig. 1A) indicate that both deterministic and stochastic

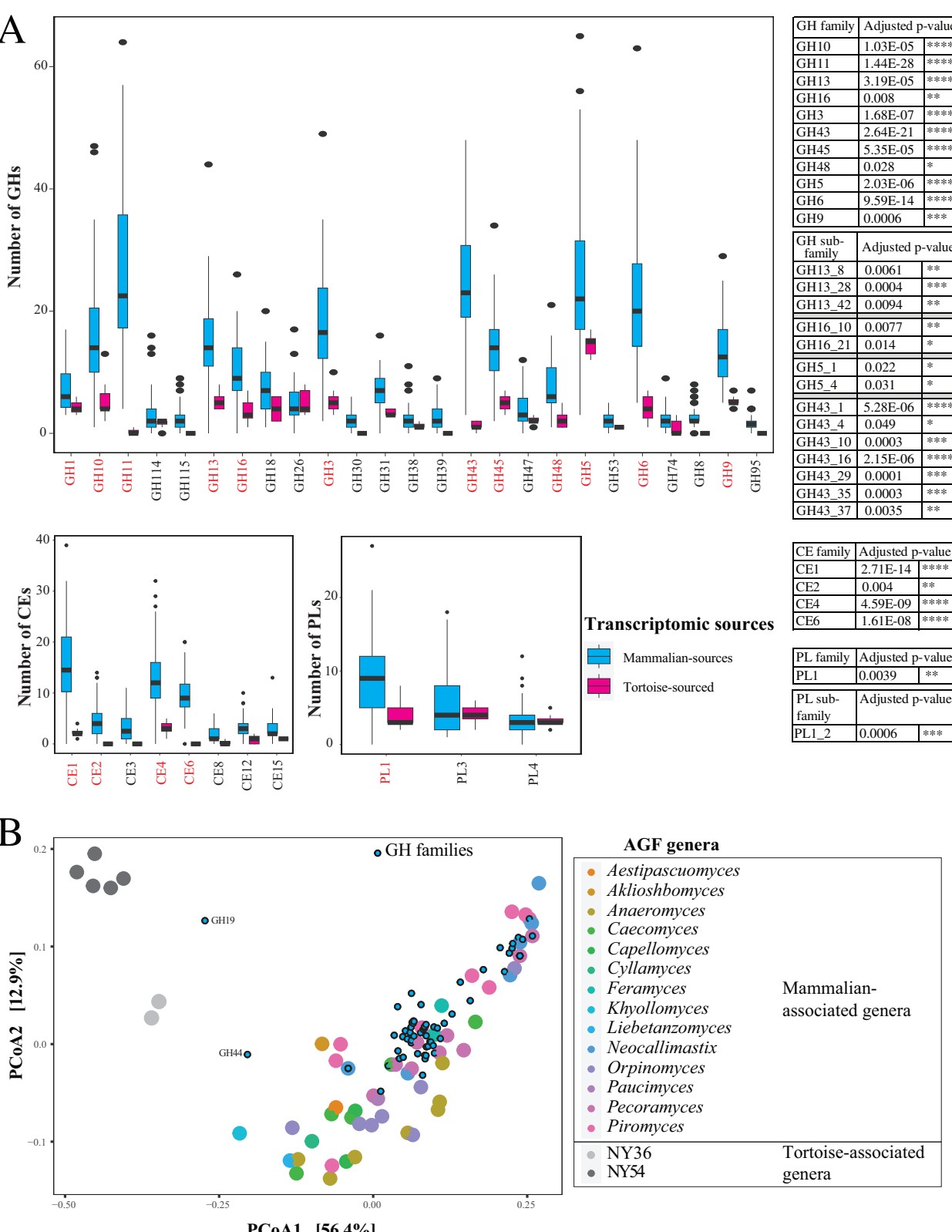

processes play a role in shaping AGF communities to different extents in various tortoise samples analyzed. In most samples (9/11) niche-based selection predominates, resulting in a community dominated by T-AGF[46]. The preference of these hosts to T-AGF could be explained by the lower temperature optima (for *Testudinimyces*) and wider temperature range (for both *Testudinimyces* and *Astrotestudinimyces*) compared to M-AGF aiding in their survival and growth in the

poikilothermic (cold-blooded) tortoises, where lower and wider variation in internal temperature prevail[33]. As well, the slower growth of representatives of the genus *Testudinimyces* mirrors the slower basal metabolic rate and the extremely long food retention time in tortoises (12–14 days)[29], allowing ample time for substrate colonization. On the other hand, the prevalence of AGF communities similar to mammalian AGF communities in some tortoises could be a reflection of a neutral

**Fig. 4 | CAZyome composition difference between tortoise-sourced and mammalian-sourced (n = 54) strains. A** Box and whisker plots for the distribution of the total number of GHs (top), CEs (bottom left), and PLs (bottom right) identified in the transcriptomes (mammalian sourced, cyan; tortoise sourced, pink). Only CAZy families with >100 total hits in the entire dataset are shown, and CAZy families that were significantly more abundant in mammalian versus tortoise transcriptomes are shown in red text. Boxplots in Fig. 1B, D, E extend from the first to the third quartile and the median is shown as a thick line in the middle. The whiskers extending on both ends represent variability outside the quartiles and are calculated as follows: Minimum whisker = minimum quartile − 1.5 × inter-quartile range; Maximum whisker = maximum quartile + 1.5 x inter-quartile range. All points outside the box and whiskers are outliers. Two-sided Wilcoxon test adjusted p values for the significance of difference in CAZyome composition for the families in red text are shown to the right, along with the values for GH5, GH13, GH16, GH43, and PL1 sub-families. **B** Principal coordinate analysis (PCoA) biplot based on the GH families composition in the studied transcriptomes. The % variance explained by the first two axes is displayed on the axes, and strains are color-coded by AGF genus, as shown in the figure legend to the right, while GH families are shown as smaller cyan spheres with black borders. Source data are provided as a Source Data file.

selection process, where community structures are independent of species traits and governed mostly by stochastic processes[46].

The reason for the stark difference in AGF community structure between the plowshare and one of the Galapagos samples compared to all other 9 samples (Fig. 1A) is currently unclear. The highly versatile feeding preferences of Plowshare and Galapagos tortoises (frugivory, graminivory, and foliovory, Table S1) and the humid island original habitats for both species are worth noting, but the role played by these factors in shaping the AGF community is unclear. It should be noted that the other Galapagos tortoise replicate studied here had a T-AGF (*Testudinimyces*)-dominated AGF community.

Admittingly, the relatively limited sampling effort in this study, as well as the fact that 10/11 samples were obtained from a single location (Oklahoma City Zoo, Oklahoma, USA), render extrapolating such observations to the global AGF community in tortoises in their natural habitats inappropriate. The bacterial community could significantly differ between zoo and wild animals[47], although whether such patterns could be extrapolated to AGF communities in mammals and tortoises is currently unclear. Therefore, sampling limitations prevent us not only from further investigating the role of niche versus neutral processes in shaping AGF community in tortoises, but also from determining whether the observed patterns of AGF community composition, structure, and diversity could be extrapolated to the broader global free-living tortoise community.

The exact ecological role and services rendered by T-AGF to their hosts, if any, are currently unclear. Given their relatively low numbers (Fig. 1E) in the ecosystem, as well as their relatively curtailed CAZyme repertoire (Fig. 4, Fig. S8, Table S3, and Dataset S4), their relative contribution to substrate depolymerization in their hosts appears minor. As well, a role in oligomer conversion to monomers, followed by monomer fermentation to absorbable volatile fatty acids, could be postulated. Alternatively, T-AGF could be rendering ecological services unrelated to food digestion in their host, e.g., modulating community and preventing pathogenic microbes' establishment via secondary metabolites production and niche competition, akin to the gut and skin microbiome role in colonization resistance in human[48,49]. Finally, the possibility that T-AGF are dispensable commensals, rather than indispensable symbionts could not be discounted, given their low loads in the tortoise GIT (Fig. 1E).

Regardless of their adaptive strategies and putative role in the tortoise GIT, our findings have important implications for our understanding of the evolutionary history of *Neocallimastigomycota*. Prior efforts based on available M-AGF taxa estimated an AGF divergence time of 67 Mya[27,32]. Such estimate is close to the K-Pg extinction event (66 Mya) and coincides with the evolution of mammalian AGF host families[50–56] and the associated evolutionary innovations in hosts' alimentary tract architecture, as well as the evolution of grasses in the family *Poaceae*[57]. Our results describe two distinct, deep-branching lineages that evolved 37–45 Mya prior to these events. The described genera hence represent the earliest known form of host-anaerobic fungal associations known to date; and demonstrate that AGF evolution predates the events previously recognized as the driving force behind forging *Neocallimastigomycota* evolution as a distinct fungal phylum. However, our results, while identifying novel deep-branching

AGF lineages with preference to tortoises over mammals, should not be interpreted as arguing for a strict T-AGF-tortoise coevolution pattern, where earliest *Amniote* or *Testudines* ancestors have acquired and maintained these T-AGF taxa. This is due to the lack of a continuous line of herbivory within these lineages as well as the obvious absence of sampling from extinct ancestors. Further, it is important to note that our results do not challenge the key role played by the rise of mammalian herbivory post the K-Pg extinction event and the evolution of mammalian families with dedicated fermentation chambers in AGF evolution. The establishment of AGF in the mammalian herbivorous gut has spurred an impressive wave of AGF family- and genus-level diversification[27] and the acquisition of genes enabling efficient cellulose and hemicellulose degradation via HGT[39] to fully utilize the newly evolved grasses (family *Poaceae*) as a primary food source[32,39]. These innovations, in turn, enabled the establishment of AGF as indispensable members of the GIT tract of herbivorous mammals[23,58–60]. Indeed, most of the AGF-identified diversity and biomass on earth currently resides in mammalian, rather than non-mammalian, herbivores.

Finally, comparative analysis between M- and T-AGF clearly indicates a significantly lower frequency of gene acquisition via HGT in T-AGF compared to M-AGF (Table 1). Our analysis suggests that a primary purpose of HGT in T-AGF is to enable their transition from an aerobic ancestor to an anaerobic lineage, a prerequisite for their establishment in the tortoise AGF tract. Only a few (13 out of 35) HGT events were associated with improving plant degradation capacity in T-AGF lineages, which is, in turn, reflected in the slower cellulose-degradation ability and the lack of xylan degradation abilities in T-AGF taxa (Fig. S7). Further, most of the HGT events identified in T-AGF in this study were also observed in M-AGF indicating ancient acquisition events prior to T-AGF and M-AGF split (Fig. 3). However, these relatively few ancient HGT events were followed by a more extensive second wave of HGT-mediated gene acquisition that occurred solely in M-AGF and was mostly responsible for equipping M-AGF with a powerful plant biomass degradation machinery enabling their propagation, establishment, and competition in the highly competitive, bacteria and archaea dominated rumen and hindgut in mammalian herbivores[32,39].

What prevented T-AGF genera from undergoing a similar massive acquisition of CAZymes to improve their competitive advantage in the tortoise GI tract and beyond? We provide two possible explanations for the observed deficiency. First, such differences could be niche-related. Mammalian rumen and hindguts are characterized by higher temperatures, larger food intake, rapid digestion and substrate turnover, higher microbiome density and diversity, and higher overall metabolic activity[61]. Such conditions provide for a more active milieu of cells, extracellular DNA, and viruses with higher opportunities for HGT through transduction, natural uptake, and transformation. Second, such differences in HGT frequency could be related to the hyphal growth pattern of AGF taxa. T-AGF genera identified appear to be polycentric, and such taxa produce a lower number of zoospores and can depend on hyphal propagation as a means of reproduction[33]. In contrast, most M-AGF genera (16/20 genera), including the earliest evolving ones

**Table 1 | HGT events identified in the two tortoise genera (B: *Astrotestudinimyces*; T: *Testudinimyces*), the affiliation of HGT donor, and distribution of the event in mammalian AGF (M-AGF)**

| COG/ KEGG Classification | Function imparted/ Pathway | Affiliation of donor | | Distribution in Tortoise genera | Occurrence In mammalian AGF | Same donor? |
|---|---|---|---|---|---|---|
| | | **Phylum/Class** | **Kingdom/ Clade** | | | |
| **Cellular Processes and Signaling** | | | | | | |
| *[O] Posttranslational modification, protein turnover, chaperones* | DsbA | Cnidaria | Metazoa | Both | Yes | No |
| **Information Storage and Processing** | | | | | | |
| *[L] Replication, recombination and repair* | methylated DNA protein-cysteine methyl transferase | Firmicutes | Bacteria | Both | Yes | Yes |
| *[J] Translation, ribosomal structure and biogenesis* | GTP-binding | Mixed phyla | Bacteria | Both | Yes | Yes |
| **Metabolism** | | | | | | |
| *[E] Amino acid transport and metabolism* | ADP-ribosyl arginine hydrolases | Firmicutes | Bacteria | T | Yes | Yes |
| | Aspartate-ammonia ligase | Bacteroidetes | Bacteria | Both | Yes | Yes |
| | Cysteine synthase | Firmicutes | Bacteria | Both | Yes | No |
| | Tryptophan synthase (trpB) | Verrucomicrobia | Bacteria | T | Yes | Yes |
| *[H] Coenzyme transport and metabolism* | dephospho CoA kinase | | Amoebozoa | T | Yes | Yes |
| | NadB (L-aspartate oxidase) | Myxococcota | Bacteria | Both | Yes | Yes |
| *[C] Energy production and conversion* | Lactate dehydrogenase and 2-hydroxyacid dehydrogenase | Firmicutes | Bacteria | Both | Yes | Yes |
| | lipoamide dehydrogenase | Mixed phyla | Bacteria | Both | Yes | Yes |
| | bifunctional aldehyde/alcoholDH family of Fe-ADH | Cyanobacteria | Bacteria | Both | Yes | Yes |
| *[F] Nucleotide transport and metabolism* | guanine deaminase | Firmicutes | Bacteria | B | No | N/A |
| | thymidine kinase | Alpha-Proteobacteria | Bacteria | T | Yes | Yes |
| | anaerobic ribonucleoside triphosphate reductase | TM6 | Bacteria | Both | Yes | Yes |
| *[Q] Secondary metabolites biosynthesis, transport and catabolism* | Dehydrogenases | Firmicutes | Bacteria | Both | Yes | Yes |
| | Dehydrogenases | Firmicutes | Bacteria | T | Yes | Yes |
| *[KO] Metabolism of terpenoids and polyketides* | Squalene-hopene cyclase | Ciliophora | Alveolata | Both | Yes | Yes |
| *[G] Carbohydrate transport and metabolism* | | | | | | |
| Glycoside Hydrolases | GH13 | Firmicutes | Bacteria | Both | Yes | Yes |
| | GH1 | Firmicutes | Bacteria | Both | Yes | Yes |
| | GH5 | Firmicutes | Bacteria | Both | Yes | Yes |
| | GH43 | Firmicutes | Bacteria | Both | Yes | Yes |
| | GH10 | Firmicutes | Bacteria | Both | Yes | Yes |
| | GH11 | Fibrobacteres | Bacteria | T | Yes | Yes |
| | GH88 | Mollicutes/ Firmicutes | Bacteria | Both | Yes | Yes |
| | GH3 | Firmicutes | Bacteria | B | Yes | Yes |
| | GH53 | Firmicutes | Bacteria | Both | No | N/A |
| | GH48 | Firmicutes and Actinobacteria | Bacteria | T | Yes | Yes |
| Carbohydrate Esterases | CE1 | Firmicutes and Fibrobacteres | Bacteria | B | Yes | Yes |
| | CE15 | Bacteroidetes | Bacteria | Both | Yes | Yes |
| Polysaccharide Lyases | PL4 | Bacteroidetes | Bacteria | Both | Yes | Yes |
| **Poorly characterized** | | | | | | |
| *[KO] Not Included in Pathway or Brite* | aminopeptidase (Peptidase MEROPS Family M18) | Firmicutes | Bacteria | Both | No | N/A |
| *No COG/KEGG* | TerD_like | Firmicutes | Bacteria | Both | No | N/A |
| | uncharacterized protein | | Metazoa | Both | No | N/A |
| | uncharacterized protein | Firmicutes | Bacteria | Both | No | N/A |

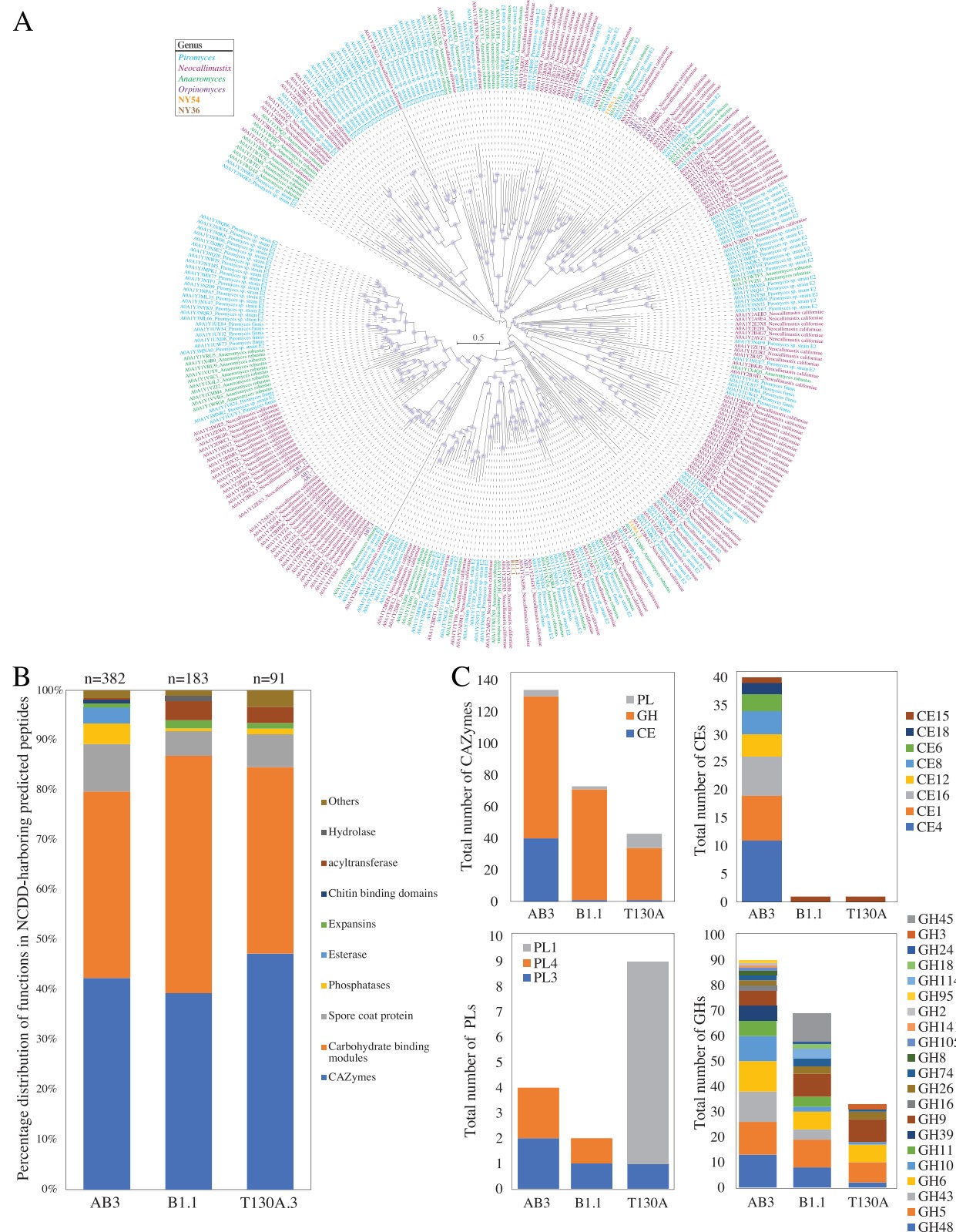

(e.g., *Khoyollomyces* and *Piromyces*), are monocentric, with strict dependency on sporangial development and free zoospore release followed by encystment and growth. Given the fact that fungal zoospores are naturally competent[62,63] and represent the most appropriate and logical stage for DNA uptake by the AGF, such differential prevalence could contribute to the observed differences in HGT between both groups.

## Methods

### Ethics statement

Fecal samples from 11 tortoises were obtained between November 2020 and March 2022 (Table S1). Fecal samples were collected post deposition by the animal. Since the process does not involve any direct interaction with tortoises, no specific ethical review was necessary. All samples were obtained by trained zoo personnel (Oklahoma City Zoo,

**Fig. 5 | Comparative cellulosomal analysis between representatives of the two tortoise-affiliated genera (genus *Astrotestudinimyces*, strain B1.1; and genus *Testudinimyces*, strain T130A) and one mammalian affiliated strain (*Orpinomyces joyonii* strain AB3). A** Maximum likelihood mid-point rooted phylogenetic tree showing the relationship between scaffoldin ScaA protein homologs identified in *Orpinomyces joyonii* strain AB3 (12 copies denoted AB3_1 through AB3_12 and shown in purple text), *Astrotestudinimyces* strain B1.1 (2 copies denoted B1.1_and B1.1_2 and shown in brown boldface text), and *Testudinimyces* strain T130A (2 copies denoted T130A_and T130A_2 and shown in orange boldface text) in comparison to a reference set of 319 Neocallimastigomycota ScaA homologs retrieved from Uniprot. All reference ScaA homologs are shown with their Uniprot ID

followed by the AGF strain name color-coded by genus, as shown in the legend. **B** Comparison of the percentage distribution of functions (as predicted by NCBI Conserved Domain database) encoded by cellulosomal peptides (all predicted peptides harboring a non-catalytic dockerin domain in the two tortoise affiliated genera (genus *Astrotestudinimyces*, strain B1.1; and genus *Testudinimyces*, strain T130A) and the mammalian affiliated strain (*Orpinomyces joyonii* strain AB3) and destined to the extracellular milieu (as predicted by DeepLoc)). The total number of peptides is shown above each column. **C** CAZyome composition of the predicted cellulosome in the three strains compared. Source data for Fig. 5**B**, **C** are provided as a Source Data file.

Oklahoma City Oklahoma), or by a farm owner (Walters Oklahoma) as part of the routine process of cleaning their habitat from deposited feces.

## Samples

Fecal samples from 11 tortoises belonging to 8 genera and 9 species were obtained between November 2020 and March 2022 (Table S1). All samples originated from animals kept at the Oklahoma City Zoo (Oklahoma City, Oklahoma, USA), except one Sulcata (African spurred) tortoise sample (*Centrochelys sulcate*), which was obtained from a local farm near Walters, OK, USA (34°28'43.3"N 98°13'33.0"W). Specimen collection from wild tortoise populations is exceedingly difficult since many of the sampled tortoise spp. are critically endangered, e.g., plowshare tortoise[64], and/or have a very limited geographic range (Table S1). Tortoises from the Oklahoma City Zoo are all housed in a separate building (the Herpetarium), secluded from mammalian habitats and enclosures. Freshly deposited samples were placed in 15- or 50-mL sterile conical centrifuge tubes and transferred on ice to the laboratory, where they were stored at −20 °C. All samples originated from individual animals and were not adulterated during sampling with dust, dirt, or feces from other subjects. All sampled tortoises were visibly healthy during the time of sampling.

## Amplicon-based diversity surveys

DNA extraction from fecal samples was conducted using DNeasy Plant Pro Kit (Qiagen Corp., Germantown, Maryland, USA) according to the manufacturer's instructions and as previously described[27]. PCR amplification targeting the D2 region of the LSU rRNA utilized the DreamTaq Green PCR Master Mix (ThermoFisher, Waltham, Massachusetts, USA), and AGF-specific primers AGF-LSU-EnvS For: 5′-GCGTTTRRCACCASTGTTGTT-3′, and AGF-LSU-EnvS Rev: 5′-GTCAA-CATCCTAAGYGTAGGTA-3′[65]. The primers target a -370 bp region of the LSU rRNA gene (corresponding to the D2 domain), hence allowing for high throughput sequencing using the Illumina MiSeq platform. Primers were modified to include the Illumina overhang adapters. PCR reactions contained 2 μl of DNA, 25 μl of the DreamTaq 2X Master Mix (Life Technologies, Carlsbad, California, USA), 2 μl of each primer (10 μM) in a 50 μl reaction mix. The PCR protocol consisted of an initial denaturation for 5 min at 95 °C followed by 40 cycles of denaturation at 95 °C for 1 min, annealing at 55 °C for 1 min and elongation at 72 °C for 1 min, and a final extension of 72 °C for 10 min. PCR products were individually cleaned to remove unannealed primers using PureLink® gel extraction kit (Life Technologies), and the clean product was used in a second PCR reaction to attach the dual indices and Illumina sequencing adapters using Nexterra XT index kit v2 (Illumina Inc., San Diego, California, USA). These second PCR products were then cleaned using PureLink® gel extraction kit (Life Technologies, Carlsbad, California, USA), individually quantified using Qubit® (Life Technologies, Carlsbad, California, USA), and pooled using the Illumina library pooling calculator (https://support.illumina.com/help/pooling-calculator/pooling-calculator.htm) to prepare 4–5 nM libraries. Pooled libraries were sequenced at the University of Oklahoma Clinical Genomics Facility (Oklahoma City, Oklahoma, USA) using the MiSeq

platform. Negative (no DNA) controls were conducted with all amplifications.

## Sequence data analysis

Forward and reverse Illumina reads were assembled using make.-contigs command in mothur[66], followed by screening to remove sequences with ambiguous bases, sequences with homopolymer stretches longer than 8 bases, and sequences that were shorter than 200 or longer than 380 bp. Sequence assignment to AGF genera was conducted using a two-tier approach as recently described[27]. Briefly, sequences were first compared by blastn to the curated D1/D2 LSU rRNA AGF database (www.anaerobicfungi.org) and were classified as their first hit taxonomy if the percentage similarity to the first hit was >96% and the two sequences were aligned over >70% of the query sequence length. For all sequences that could not be confidently assigned to an AGF genus by blastn, insertion into a reference LSU tree (with representatives from all cultured and uncultured AGF genera and candidate genera) was used to assess novelty. These genus-level assignments were then used to build a taxonomy file in mothur, which was subsequently used to build a shared file using the mothur commands phylotype and make.shared. The genus-level shared file was used for all downstream analyses, as detailed below.

For comparative diversity analyses, the AGF genus-level shared file from the 11 tortoise samples studied here was combined with an AGF genus-level shared file from a subset of mammalian samples generated in a recent prior study conducted by our research group[27]. Fecal samples in both studies were obtained using the same sampling procedures and were subjected to identical DNA extraction, amplification, sequencing, and sequence analysis procedures. The subset of mammalian AGF hosts included a comparable size from each of the five most commonly sampled and numerous mammalian hosts: cattle ($n = 25$) (*Bos taurus*), sheep ($n = 25$) (*Ovis aries*), goats ($n = 25$) (*Capra hircus*), white-tail deer ($n = 24$) (*Odocoileus virginianus*), and horses ($n = 25$) (*Equus caballus*) (Dataset S2A). The combined AGF genus-level shared file was used in the R package phyloseq[67]. Another mammalian subset ($n = 11$; Dataset S2B) comprised of animals housed in the Oklahoma City Zoo was also compared to the 11 tortoise samples to normalize for any effects the domestication status might have had on diversity results. Alpha diversity estimates (observed number of genera, Shannon, Simpson, and Inverse Simpson diversity indices) were calculated using the command estimate_richness. To compare beta diversity and community structure between AGF communities in tortoises and AGF communities in canonical mammalian hosts, we used the ordinate command in the phyloseq R package to calculate weighted Unifrac beta diversity indices and used the obtained pairwise values to construct ordination plots (both PCoA and DPCoA) using the function plot_ordination in the phyloseq R package.

## Quantitative PCR

We quantified total AGF load in the 11 tortoise samples and compared it to the AGF load in 10 cattle, 10 goats, 10 sheep, and 10 horses (sample names in red text in Dataset S2A) using quantitative PCR. The same primer pair (AGF-LSU-EnvS and AGF-LSU-EnvS Rev) used in the

amplicon-based diversity survey described above was also used for qPCR quantification. The 25-μl PCR reaction volume contained 1 μl of extracted DNA, 0.3 μM of primers AGF-LSU-EnvS primer pair, and SYBR GreenER™ qPCR SuperMix for iCycler™ (ThermoFisher, Waltham, Massachusetts, USA). Reactions were run on a MyiQ thermocycler (Bio-Rad Laboratories, Hercules, California, USA). The reactions were heated at 95 °C for 8.5 min, followed by 40 cycles, with one cycle consisting of 15 s at 95 °C and 1 min at 55 °C. A pCR 4-TOPO or pCR-XL-2-TOPO plasmid (ThermoFisher, Waltham, Massachusetts, USA) containing an insert spanning ITS1-5.8S rRNA-ITS2-D1/D2 region of 28S rRNA from a pure culture strain was used as a positive control, as well as to generate a standard curve. The efficiency of the amplification of standards (E) was calculated from the slope of the standard curve and was found to be 0.89.

### Isolation of AGF from Tortoises

Isolation of AGF from fecal samples of tortoises was conducted using established enrichment and isolation procedures in our laboratory[25,33]. A sequence-guided strategy, where samples with the highest proportion of novel, yet-uncultured AGF taxa were prioritized, was employed. To account for the poikilothermic (ectothermic) nature of the host, and the fact that the tortoise gut community is often exposed to variable temperatures, we enriched for T-AGF at a range of temperatures (30 °C, 39 °C, and 42 °C). Finally, the rumen fluid medium used for enrichment and isolation was amended with cellobiose (RFC medium) in addition to an insoluble substrate (switchgrass) and antibiotics (50 μg ml⁻¹ chloramphenicol, 20 μg ml⁻¹ streptomycin, 50 μg ml⁻¹ penicillin, 50 μg ml⁻¹ kanamycin, and 50 μg ml⁻¹ norfloxacin).

### Transcriptomic sequencing

Transcriptomic sequencing of 8 representative T-AGF isolates was conducted as described previously[27,32]. Briefly, biomass from cultures grown in RFC medium was vacuum filtered and used for total RNA extraction using an Epicentre MasterPure Yeast RNA purification kit (Epicentre, Madison, Wisconsin, USA) according to the manufacturer's instructions. RNA-seq was conducted on an Illumina HiSeq2500 platform using 2 × 150 bp paired-end library at the Oklahoma State University Genomics and Proteomics Core Facility (Stillwater, Oklahoma, USA). RNA-seq reads were quality trimmed and de novo assembled using Trinity (v2.14.0) and default parameters. Assembled transcripts were clustered using CD-HIT[68] (identity parameter of 95% (−c 0.95)) to identify unigenes. Following, peptide and coding sequence prediction was conducted on the unigenes using TransDecoder (v5.0.2) with a minimum peptide length of 100 amino acids (https://github.com/TransDecoder/TransDecoder). BUSCO[69] was used to assess transcriptome completeness using the fungi_odb10 dataset (modified to remove 155 mitochondrial protein families as previously suggested[22] (Table S4).

### Phylogenomic analysis and molecular dating

Phylogenomic analysis was conducted as previously described[27,70] using the 8 transcriptomic datasets generated in this study, as well as 52 transcriptomic datasets from 14 AGF genera previously generated by our group[27,32,39], and others[22,43,71,72], in addition to 5 outgroup Chytridiomycota genomes (Chytriomyces sp. strain MP 71, Entophlyctis helioformis JEL805, Gaertneriomyces semiglobifer Barr 43, Gonapodya prolifera JEL478, and Rhizoclosmatium globosum JEL800) to provide calibration points. The final alignment file included 88 genes that were gap-free and comprised a total of 37,044 nucleotide sites. This refined alignment was further grouped into 20 partitions, each assigned with an independent substitution model, suggested by a greedy search using PartitionFinder v2.1.1. All partition files, along with their corresponding models, were imported into BEAUti v1.10.4 for conducting Bayesian and molecular dating analyses. Calibration priors were set as previously described[32], including a direct fossil record of

Chytridiomycota from the Rhynie Chert (407 Mya) and the emergence time of Chytridiomycota (573 to 770 Mya as 95% HPD). The Birth-Death incomplete sampling tree model was employed for interspecies relationship analyses. Unlinked strict clock models were used for each partition independently. Three independent runs were performed for 30 million generations each. Tracer v1.7.1[73] was used to confirm that sufficient effective sample size (ESS > 200) was obtained after the default burn-in (10%). The maximum clade credibility (MCC) tree was compiled using TreeAnnotator v1.10.4[74].

### Transcriptomic gene content analysis and comparative transcriptomics

Transcriptomic datasets obtained from seven of the tortoise AGF isolates were compared to the 52 previously generated transcriptomic datasets from mammalian AGF isolates[22,27,32,39,43,71,72]. Gene content comparison was conducted via classification of the predicted peptides against COG, KOG, GO, and KEGG classification schemes, as well as prediction of the overall CAZyme content. COG and KOG classifications were carried out via blastp comparisons of the predicted peptides against the most updated databases downloaded from NCBI ftp server (https://ftp.ncbi.nih.gov/pub/COG/COG2020/data/ for COG 2020 database update, and https://ftp.ncbi.nih.gov/pub/COG/KOG/ for KOG database). GO annotations were obtained by first running blastp comparisons of the predicted peptides against the SwissProt database. The first SwissProt hit of each peptide was then linked to a GO number by awk searching the file idmapping_selected.tab available from the Uniprot ftp server (https://ftp.uniprot.org/pub/databases/uniprot/current_release/knowledgebase/idmapping/idmapping_selected.tab.gz). GO numbers corresponding to the first hits were then linked to their GO aspect (one of: molecular function, cellular component, or biological process) by awk searching the file "goa_uniprot_all.gaf" available from GOA FTP site (ftp://ftp.ebi.ac.uk/pub/databases/GO/goa/UNIPROT). KEGG classification was conducted by running GhostKOALA[75] search on the predicted peptides. The overall CAZyme content was predicted using run_dbcan4 (https://github.com/linnabrown/run_dbcan), the standalone tool of the dbCAN3 server (http://bcb.unl.edu/dbCAN2/) to identify GHs, PLs, CEs, AAs, and CBMs in the transcriptomic datasets.

To identify predicted functions that are unique to tortoise-associated or mammalian-associated AGF, predicted peptides from all 59 transcriptomes were compared in an all versus all blastp followed by MCL clustering. Clusters obtained were then examined to identify these clusters that are unique to both tortoise isolates, unique to one of them, or present in mammalian-associated genera but absent from both tortoise genera (thereafter "GroupD" clusters). KEGG classifications of predicted peptides belonging to each of these groups were then compared.

### Quantifying horizontal gene transfer (HGT)

We implemented an HGT detection pipeline that was previously developed and extensively validated[39] to identify patterns of HGT in T-AGF transcriptomic datasets. The pipeline involved a combination of BLAST similarity searches against UniProt databases (downloaded January 2023), comparative similarity index (HGT index, $h_U$), and phylogenetic analyses to identify potential HGT candidates. The downloaded Uniprot databases encompassed Bacteria, Archaea, Viruses, Viridiplantae, Opisthokonta-Chaonoflagellida, Opisthokonta-Metazoa, the Opisthokonta-Nucleariidae and Fonticula group, all other Opisthokonta, and all other non-Opisthokonta, non-Viridiplantae Eukaryota. Each predicted peptide from tortoise isolates transcriptomic datasets were searched against each of these databases, as well as against the Opisthokonta-Fungi (without Neocallimastigomycota representatives). Candidates with a blastp bit-score against a nonfungal database that was >100 and an HGT index $h_U$ that was ≥30 were further evaluated via phylogenetic analysis to confirm HGT

occurrence and to determine the potential donor. All potential candidates were first clustered using CD-HIT and a 95% similarity cutoff. Representatives of each cluster were then queried against the nr database using web blastp once against the full nr database and once against the *Fungi* (taxonomy ID 4751) excluding the *Neocallimastigomycetes* (taxonomy ID 451455) with an $E$ value below $e^{-10}$. The first 100 hits obtained using these two blastp searches were downloaded and combined in one FASTA file that was then combined with the AGF representative sequences and aligned using MAFFT multiple sequence aligner, and the alignment was subsequently used to construct maximum likelihood phylogenetic trees using FastTree. At this level, candidates that showed a nested phylogenetic affiliation that was incongruent to organismal phylogeny with strong bootstrap supports were deemed horizontally transferred.

Multiple safeguards were taken to ensure that HGT events reported here are not due to bacterial contamination of AGF transcripts. These included: 1. Application of antibiotics in all culturing procedures to guard against bacterial contamination. 2. The utilization of transcriptomes rather than genomes, which selects for eukaryotic polyadenylated (poly-A) transcripts prior to RNA-seq as a built-in safeguard against possible prokaryotic contamination. 3. Applying a threshold where only transcripts identified in >50% of transcriptomic assemblies from a specific genus ($n = 5$ for *Testudinimyces* and 2 for *Astrotestudinimyces*) are included. In addition, recent studies have demonstrated that GenBank-deposited reference genomes of multicellular organisms could be plagued by prokaryotic contamination[76,77]. The occurrence of prokaryotic contamination in reference donors' genomes/transcriptomes could lead to false positive HGT identification. To guard against the possibility of contamination in reference datasets, sequence data from potential donor reference organisms were queried using blastp, and their congruence with organismal phylogeny was considered a prerequisite for the inclusion of an HGT event.

### Predicted secretome in transcriptomic datasets

To identify the predicted secretome, DeepLoc 2.0[78] was used to predict the subcellular location of all predicted peptides from the transcriptomes of a representative of two different T-AGF genera (strain T130A.3 and B1.1), as well as one representative of mammalian-associated AGF genera (*Orpinomyces joyonii* strain AB3). All transcripts encoding peptides predicted to be extracellular (henceforth predicted secretome) were then subjected to run_dbcan4 (https://github.com/linnabrown/run_dbcan) to identify GHs, PLs, and CEs in the predicted secretome. In addition, the predicted secretome was searched for the presence of scaffoldin homologs via blastp comparison against a scaffoldin database (319 proteins downloaded June 2023 from Uniprot and created by searching the UniprotKB for scaffoldin and filtering the output by taxonomy using taxid Neocallimastigomycetes [451455]). Finally, the predicted secretome was also searched for the presence of NCDD via the NCBI Batch CD-search online tool and identifying the predicted peptides with hits to the CBM_10 pfam02013. All predicted extracellular peptides with NCDD were further subjected to run_dbcan4 to identify co-existing GH, PL, and CE domains.

### Identification of hydrogenosomal proteins in representatives of T-AGF

In AGF, mitochondria have been reduced to a structure lacking a mitochondrial genome called a hydrogenosome[34–36]. Hydrogenosomes in AGF are involved mainly in hydrogen production and substrate-level phosphorylation. We sought to compare the predicted hydrogenosomal content of one representative of each T-AGF genus (strains T130A.3 and B1.1), as well as one representative of mammalian-associated AGF genera (*Orpinomyces joyonii* strain AB3). All predicted peptides from the transcriptomes of the three strains predicted by DeepLoc 2.0[78] to be mitochondrial were further classified against

KEGG (by running GhostKOALA[75]) and the conserved domain database (CDD[79]).

### Proteomics sequencing and analysis

In addition to secretome prediction from transcriptomic datasets, we conducted proteomic analysis on the same T-AGF described above (*Testudinimyces gracilis* strains T130A.3 and *Astrotestudinimyces divisus* strains B1.1). Proteomic analysis was conducted on two fractions: biomass, and cellulose-bound. Briefly, cultures were grown in RFC media until mid-exponential phase (typically 3 days for *Astrotestudinimyces* strain B1.1, and 7 days for *Testudinimyces* strain T130A). Biomass fraction was first collected by centrifugation ($3220 \times g$ for 10 min at 4 °C). The cellulosomal fraction (in the supernatant) was separated using cellulose precipitation, as previously described[43]. Briefly, the supernatant pH was adjusted to 7.5, followed by adding cellulose (Sigmacell type 50) (0.4% w/v) and gently stirring at 4 °C for 2 h. Low-speed centrifugation ($3220 \times g$ for 10 min at 4 °C) was then used to separate the cellulosomal (pellet) fraction. Proteins bound to cellulose in the cellulosomal fraction (pellet) were then eluted in water by agitation at room temperature for 1 h, followed by removal of cellulose by centrifugation. The soluble eluates were collected by centrifugation, and frozen at −80 °C. For both fractions (biomass and cellulose-bound), the frozen samples were dried by vacuum centrifugation. Dried samples were redissolved for 30 min at RT in reducing buffered guanidine (6 M guanidine HCl, 0.1 M Tris HCl, tris(2-carboxyethyl)phosphine, pH 8.5). Debris were removed by centrifugation, and the solutes were alkylated by adding iodoacetamide to 10 mM and incubation for 30 min in the dark at RT. The alkylation reactions were then digested with trypsin using a filter-aided sample preparation (FASP) approach[80]. For FASP, the samples were loaded into 30-kDa spin filter devices (Sigma®) and subjected to three buffer exchanges into 8 M urea, 0.1 M TrisHCl, pH 8.5, followed by three additional buffer exchanges into digestion buffer (100 mM TrisHCl, pH 8.5). For the final buffer exchange, samples were concentrated to ~10 μl in digestion buffer, followed by dilution with 75 μl of digestion buffer containing 0.75 μg of trypsin/LysC mix (Promega). Reactions were digested overnight at 37 °C, and the trypsinolysis products were recovered by centrifugation of the FASP device. Recovered peptides were desalted using centrifugal devices loaded with C18 resin following the manufacturer's recommendations (HMMS18R, The Nest Group). The desalted peptides were frozen and dried by vacuum centrifugation and redissolved in 0.1% aqueous formic acid immediately prior to analysis by LC-MS/MS.

For LC-MS/MS, peptides were injected onto a 75 μm × 50 cm nano-HPLC column packed with 1.9-micron C18 beads (Thermo PN 164942) connected to an Easy-nLC 120 nano-HPLC system configured for two-column vented trap operation. Peptides were separated by gradient chromatography using 0.1% aqueous formic acid as mobile phase A and 80:20:0.1 acetonitrile/water/formic acid as mobile phase B. Peptides separations used a gradient of 4–32% mobile phase B delivered over a period of 120 min. Eluted peptides were ionized in a Nanospray Flex Ion source using stainless steel emitters (Thermo). Peptide ions were analyzed in a quadrupole-Orbitrap mass spectrometer (Fusion model, Thermo) using a "high/low" "top-speed" data-dependent MS/MS scan cycle that consisted of an MS1 scan in the Orbitrap sector, ion selection in the quadrupole sector, high energy collision in the ion routing multipole, and fragment ion analyses in the ion trap sector. The details regarding the MS programming are provided (Table S5).

RAW files from the mass spectrometer were searched against the corresponding transcriptome predicted peptides database using the MaxQuant application (v2.0.2.0[81]). Searches utilized MaxQuant defaults, supplemented with two additional peptide modifications: deamidation of N/Q residues, and Q cyclization to pyroglutamate. The MaxQuant "match between runs" algorithm was not used. Sequences for reversed-sequence decoy proteins and common contaminants

proteins were utilized for the database searches but were removed from the final MaxQuant protein results.

## Statistics and reproducibility
No statistical method was used to predetermine sample size. No data were excluded from the analyses. The experiments were not randomized. The Investigators were not blinded to allocation during experiments and outcome assessment.

## Reporting summary
Further information on research design is available in the Nature Portfolio Reporting Summary linked to this article.

## Data availability
Illumina amplicon reads generated in this study have been deposited in GenBank under BioProject accession number PRJNA997953 (https://www.ncbi.nlm.nih.gov/bioproject/997953), and BioSample accession numbers SAMN36694530- SAMN36694536 (https://www.ncbi.nlm.nih.gov/biosample?LinkName=bioproject_biosample_all&from_uid=997953). RNA-seq reads from tortoise isolates have been deposited in GenBank under BioProject accession number PRJNA997953, and Bio-Sample accession numbers SAMN36694608- SAMN36694614 (https://www.ncbi.nlm.nih.gov/biosample?LinkName=bioproject_biosample_all&from_uid=997953). Source data are provided with this paper.

## Code availability
Code for phylogenomic analysis is available at https://github.com/stajichlab/PHYling_unified. Code used to create other figures is available at https://github.com/nohayoussef/AGF_Tortoises and is deposited in Zenodo[82].

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

## Acknowledgements

We thank Drs. Jennifer D'Agostino and Rebecca Snyder (Oklahoma City Zoo) for collecting and providing tortoise fecal samples. This work has been supported by the NSF grant number 2029478 to M.S.E. and N.H.Y. Computational resources used at Oklahoma State University were supported by NSF grant OAC-1531128.

## Author contributions

Conceptualization: M.S.E., N.H.Y., C.J.P.; Methodology: M.S.E., N.H.Y., C.J.P., Y.W., S.H., J.R.; Formal analysis: N.H.Y., C.J.P., Y.W.; Investigation: C.J.P., D.K.J., E.E.E., A.L.J., C.H.M.; Resources: M.S.E., N.H.Y.; Data Curation: N.H.Y.; Writing—Original Draft: M.S.E., N.H.Y.; Writing—Review & Editing: M.S.E., N.H.Y., C.J.P., E.E.E., C.H.M., Y.W.; Visualization: N.H.Y., C.J.P., Y.W.; Supervision: N.H.Y., M.S.E.; Project administration: N.H.Y., M.S.E.; Funding acquisition: N.H.Y., M.S.E.

## Competing interests

The authors declare no competing interest.
