## [Peer Review File · Nature Communications]

Anaerobic fungi in the tortoise alimentary tract illuminate early stages of host-fungal symbiosis and *Neocallimastigomycota* evolutionREVIEWER COMMENTS

Reviewer #1 (Remarks to the Author):

What are the noteworthy results? Will the work be of significance to the field and related fields? How does it compare to the established literature? If the work is not original, please provide relevant references.

The results in this manuscript, showing that tortoises harbour a low diversity of divergent anaerobic fungi are intriguing.

The work is original and it extends understanding of microbial contributions to anaerobic nutrition more generally from mammals to terrestrial, herbivorous vertebrates. The anaerobic fungi must logically have had an evolutionary history that predates the diversification of mammals because their phylogenetic sister group of aerobic fungi is much older than mammals. This manuscript raises the possibility that the fungi may have colonized any anaerobic, herbivore guts of, for example, herbivorous dinosaurs that were available through geological time.

However, the interpretation that tortoise anaerobic fungi represent a snapshot of what ancestral anaerobic fungi might have been, before ancestors of mammals and tortoises diverged is unconvincing because it assumes host-fungus coevolution rather than the more likely possibility that the anaerobic fungi are not host specific, colonizing anaerobic guts relatively quickly wherever they occur. Fig. 1 argues for frequent host jumping between mammals and tortoises because they share the same fungi, and the most common fungus to occur in tortoises also occurs commonly in mammals.

Does the work support the conclusions and claims, or is additional evidence needed?

1. The results are consistent with recent and repeated colonization of tortoise guts by anaerobic fungi that are common in mammals and vice versa rather than with maintenance of genera of anaerobic fungi starting with early amniote evolution and continuing to modern tortoises. Comparing Figs. 1A and 1B, it looks like fungus NY54 is most common in tortoises and it is also occurs widely across mammals. NY56 and NY36 predominated in one tortoise each, and appeared sporadically in mammals. And two tortoises had what appear to be a population of typical mammalian anaerobic fungi.

2. Line 65. Tortoises are herbivores with digestive systems that can accommodate anaerobic fungi, but to argue that the anaerobic fungi have been in turtle digestive tracts for longer than the age of the Testudinidae, it would be important to present evidence that ancestral turtles were herbivores. Even if SOME extinct turtles were herbivores, it would be important for the authors to provide evidence against the possibility that ancestral turtle lineages should be reconstructed as aquatic carnivores. Within turtles, the tortoise lineage is only about 40 Myr old. The recent relatively origin of the tortoises suggests that their pool of anaerobic fungi are not ancestral, but the result of secondary colonization.

3. The absence of evidence of herbivory as the ancestral state through all of the ancestors connecting tortoises with the first turtles is a problem for this argument: Phylogenomic and molecular clock timing analysis estimated an evolutionary time of 112 Mya, 480 and 104 Mya for the genera *Astrotestudinomyces*, and *Testudinomyces*, respectively. Such 481 estimates predate the evolution of all current mammalian families known to harbor AGF (59-65). 482 More importantly, it coincides with an increased diversification in fossil records of extinct turtle 483 lineages during the lower Cretaceous (145-100 Mya)

Are there any flaws in the data analysis, interpretation and conclusions? Do these prohibit publication or require revision

4. The study does not control for sampling depth, and the substantial editing needed to take this into account would result in a much different manuscript. Fungi from 11 tortoises in a zoo (this study) are being compared with fungi from many studies from varied mammals. This could explain low diversity in poorly sampled tortoises vs high diversity well sampled mammals.

5. Due to the sampling problem, the following are unsupported:

346 Assessment of alpha diversity patterns indicated that tortoises harbored a significantly
347 less diverse AGF community when compared to placental mammals (p-value <0.04) (Figure
2a).

Fig. 1B and 2A show low diversity of anaerobic fungi in 8 tortoise fecal samples but alpha diversity in two of the tortoises was as high or higher than for mammals (Fig. 2a) and in a third, it was about as high as in many mammals.

6. It is not reasonable to attribute the difference to host class when sampling of mammals and tortoises was completely different:

Host class (Mammalia versus Reptilia) explained 47.6% of the variance (adonis p-351 value = 0.001)

7. The collective predominance and ubiquity of these genera is in stark contrast to their
478 rarity and low relative abundance, when encountered, in mammalian hosts (Figure 1b, (27)).

Is the methodology sound? Does the work meet the expected standards in your field?

The methods section is mostly complete. However, a couple of points need to be resolved.

8. The issue of possible contamination must be addressed. No data are provided to document absence of contamination. Using only transcripts, distinguishing between genes that are from contaminating bacteria embedded in the fungi or closely adhering to the fungi vs genes that are horizontally transferred from bacteria to become part of the tortoise genome is difficult.

9. More information is needed about the quality of mRNA libraries from the tortoises. Authors mention that BUSCO transcripts were used to assess mRNA completeness, but (unless I missed them) they do not present the results. BUSCO are conserved and presumably highly transcribed genes that may not provide a good sense of diversity of CAZy mRNA recovered.

10. Statistics, language and analysis

- Line 348: "Community assessment using PCoA constructed using phylogenetic similarity-based weighted Unifrac..." wording is confusing
- Line 352: Does the DPCoA show that NY36, NY54, NY56 and the paucity of all other AGF is responsible for community structure distinction?
- Line 599: "Results of two-tailed ANOVA..." incomplete sentence in figure legend. Should include that significance is shown above the boxplots, and the test that was used (t-tests? ANOVA isn't used for pairwise comparisons), as in Fig 1E. Also include what asterisks mean.
- Line 603: "precentage" typo
- In Figure 2A: Multiple pairwise t-tests increases chances of type I errors. Consider using post hoc analysis (e.g. Tukey's test).
- Consider including what steps were taken to guard against detection of false HGT events due to bacterial contamination
- Fig 2D vs Fig 4B: why is 2D a DPCoA whereas 4B is a PCoA?

Other points.

Use appropriate cladistic language to avoid incorrect interpretations of evolutionary history.

11. *Khoyollomyces* is no earlier evolving than any of the other anaerobic fungi because all originated from a common ancestral species that existed at a point in geological time. It would be correct to say that the most ancient divergence in *Neocallimastigomycetes* was between *Khoyollomyces* and all other known anaerobic taxa.

genus *Khoyollomyces* recently identified as 339 the earliest evolving AGF genus (55)

12. When giving an age of a group, it is essential to specify which event in the group's origin is being dated. So was the earliest phylogenetic divergence among the aerobic fungi of tortoises 112.19 Ma, or was the divergence of the lineage of tortoise anaerobic fungi from the lineage of mammalian anaerobic fungi 112.19 Ma?

13. Fig. 1A. Scale lacks units. Should be in millions of years.

14. Fig. 1a. Would better match order of presentation if legend were flipped to show % abundance on left, AGF genera on right.

15. Fig. 1b, right panel. Are the y-axis scales correct?

Reviewer #2 (Remarks to the Author):

Summary: Anaerobic gut fungi (AGF) are a unique group of fungi that help breakdown indigestible plant fibers into nourishing food for their mammalian hosts, e.g., ruminants and hind-gut fermenters. A recent study by Meili and colleagues in 2023, which is also cited as Ref. 27, found that the AGF in mammals (M-AGF) started to diversify about 67 million years ago. Here, the same authors used metagenomic sequencing, culturing, and transcriptome profiling to discover new AGF species in the gut of tortoises. They first measured the AGF community composition and abundance from tortoise feces using rRNA sequencing (Figures 1-2). Excitingly, their analysis revealed that these new AGF species in tortoises, known as T-AGF (NY54, NY56, NY36), are an early branch of AGF that is distinct from the mammalian AGF (M-AGF). This discovery has the potential to provide fresh insights into how AGF and their animal hosts have evolved together.

The authors then isolated and cultured NY54 and NY36 from tortoise feces, and measured the expressed gene content in these T-AGF using RNA-seq. These data were used to find and align 88 conserved genes across M-AGF, T-AGF, and distant chytrid fungi to build a time-calibrated phylogenetic tree (Figure 3). This robust phylogeny confirms that T-AGF are an early-diverging lineage that diverged about 104 MYA from the common AGF ancestor shared with M-AGF. The authors then used this transcriptome dataset to show there are fewer T-AGF enzymes associated with cellulose and hemicellulose metabolism relative to M-AGF (Figures 4-5). Thus, T-AGF appear to be less capable of plant carbohydrate degradation compared to M-AGFs due to decreased CAZyme presence and reduced growth on specific plant carbohydrate media. Last, the authors identify 35 potential genes that arrived via horizontal gene transfer (HGT) into T-AGF to impart new functions (Table 1). Most of these HGT are shared with M-AGF suggesting that they were present in the ancestor of both lineages, and many of their functions are associated with carbohydrate catabolism, fermentation, and anaerobic metabolism.

This is an important manuscript because it provides solid evidence that the symbiotic association of AGF with herbivorous animals is more ancestral than previously thought. This paper will be of keen interest to general readers such as microbiologists, evolutionary biologists, and researchers interested in fungal-animal symbiosis. This manuscript will catalyze future research to uncover AGF in other herbivores and elucidate the role of early-diverging AGF in plant digestion within non-mammalian species.

Comments:

** The AGF from Ploughshare and Galapagos Zoo_604 were strikingly different from other tortoise samples and closer to M-AGF (*Anaeromyces*, *Neocallimastix*, *Orpinomyces*, etc.). This raises questions about whether these AGF were obtained from tortoise feces without any external contamination. It also makes us wonder if these particular tortoises were sick or might have had unusual digestive abilities when it comes to breaking down plant carbohydrates. We were hoping for a more in-depth discussion regarding this distinct bimodal difference in the composition of T-AGF communities.

** The authors might consider including data showing that T-AGF are unable to grow efficiently on plant carbohydrates as M-AGF in Figure 4 because this would complement and reinforce the CAZyme analysis.

** Table 1 shows that the ancestor of T-AGF and M-AGF acquired new genes through horizontal transfer that likely played a role in carbohydrate catabolism, fermentation, and adaptation to an anaerobic gut environment. Do T-AGF have intact mitochondria, degraded mitochondria, or hydrogenosomes? This type of data could provide important evidence on the timeline of AGF adaptations to the herbivore gut.

Minor comments:

** Figure 2C. The color for tortoise data points is hard to see at the zone of overlap with mammalian data points. Can tortoise points be highlighted in a way to distinguish them from the other data? E.g., dark circle around tortoise points, bring them to the foreground, no outline for mammalian points.

** Line 165: Figure 3 contains 8 T-AGF species, including *T. gracilis* sp. K01. However, the sentence says 7 species and omits K01? The Materials & Methods continue to say 7 transcriptomic datasets. Is Figure 3 wrong, or is Materials & Methods wrong?

** Line 186-187: The authors state that the "final alignment file included 88 genes ... and comprising more than 150 nucleotide sites." Did the authors mean to say 150 nucleotide sites per gene?

** Line 360: Missing "c" in -myces of *Astrotestudinimycetes*.

** Line 580: There should be a period after timetree.org before the next sentence

Reviewer #3 (Remarks to the Author):

The manuscript describes the anaerobic gut fungi (AGF) of tortoises and compares them to those found in mammalian hosts. The work uses amplicon sequencing, transcriptomics, proteomics, and phylogenomics to analyze fecal samples from several tortoises and concludes that the predominant AGF in tortoises are distinct from those predominant in mammals, and likely diverged some time before the massive evolutionary diversification of mammalian herbivores following the late Cretaceous extinctions. It is also concluded that tortoise AGF have fewer CAZymes and less horizontal gene transfer (HGT) than seen in mammalian-host AGF, and some interesting speculations are made as to why.

The work expands what is known about AGF from mammalian to non-mammalian herbivores (tortoises). In doing so it provides insights about the evolution of animal-fungal symbioses as they relate to periods of extinction and/or animal species diversification. It also expands functional and ecological knowledge of Neocallimastigomycota, an interesting phylum of early-diverging fungi which are not well understood.

The manuscript is well written and the data and methods presented appear to provide enough detail to reproduce the work, and support the claims and conclusions sufficiently well.

Minor comments:

- In Fig 1a, many of the tortoises have surprisingly low AGF diversity. Most of the tortoises are from the same zoo, so could the environment and factors such as shared pens or feedstocks be influencing AGF diversity compared to animals in the wild? Why are a few tortoise's AGFs extremely diverse in comparison? I understand from the manuscript that studying wild tortoises is difficult, but discussion of caveats would be helpful.

- Also in Fig 1a: please explain the branch lengths, or omit scale bar since it apparently was an externally downloaded tree. Also "compoisition" is misspelled. Fig 1a doesn't really show 'columns' as indicated in the legend, rather 'bars' since they are sideways.
- Table S1 would be easier to follow if the organism names matched the tree in Fig 1a were shown.
- 387, 408, 412: Change "Such pattern..." to "Such a pattern" or "This pattern"
- 427-430: These statements need references
- 469: I don't see a Figure S7 in the supplementary. It only goes up to S6.
- 552-554: Needs references
- 556-559: Needs references

Response to reviewers' comments on Nature Communications NCOMMS-23-42309

Reviewer #1 (Remarks to the Author):

The results in this manuscript, showing that tortoises harbor a low diversity of divergent anaerobic fungi are intriguing.

The work is original and it extends understanding of microbial contributions to anaerobic nutrition more generally from mammals to terrestrial, herbivorous vertebrates. The anaerobic fungi must logically have had an evolutionary history that predates the diversification of mammals because their phylogenetic sister group of aerobic fungi is much older than mammals. This manuscript raises the possibility that the fungi may have colonized any anaerobic, herbivore guts of, for example, herbivorous dinosaurs that were available through geological time.

Thank you so much for the accurate and well-articulated summary!

1. However, the interpretation that tortoise anaerobic fungi represent a snapshot of what ancestral anaerobic fungi might have been, before ancestors of mammals and tortoises diverged is unconvincing because it assumes host-fungus coevolution rather than the more likely possibility that the anaerobic fungi are not host specific, colonizing anaerobic guts relatively quickly wherever they occur. Fig. 1 argues for frequent host jumping between mammals and tortoises because they share the same fungi, and the most common fungus to occur in tortoises also occurs commonly in mammals.

Thank you. We agree with the reviewer that tortoise anaerobic fungi *DO NOT* represent a snapshot of what ancestral anaerobic fungi might have been before ancestors of mammals and tortoises diverged. The reviewer is correct that we have no definite evidence for a strict tortoise host-fungus co-evolution model. As well, the reviewer is correct in pointing out that it is currently unclear whether an unbroken chain of herbivory exists in the entire order *Testudines*, and whether the *Testudines* ancestor was an herbivore.

We also agree regarding the probability of occurrence of host jumping/secondary colonization between mammalian and tortoise hosts. However, we posit that anaerobic fungi, while capable of host jumping, could not be accurately described as completely non-host specific. The assumptions that AGF community structure is governed by stochastic processes (e.g., birth, death, colonization, priority effects) and that all AGF genera are ecologically and functionally equivalent do not satisfactorily describe the AGF community patterns observed in this study. The Tortoise-associated AGF genera (T-AGF) NY54, NY36, NY56 were present in higher relative abundance (percentage of the community) in most, but not all, tortoise samples examined (Figure 1a) and could readily be isolated from these samples. Compare that to a recent AGF community survey conducted by our group that examined AGF in 661 mammalian samples (PMID: 37365172, also see Figure 1b). In this mammalian study (conducted using the same sampling, amplification, sequencing, and analysis procedures as this tortoise study), the relative abundance of these three AGF genera (when encountered) was exceedingly minor. This pattern argues for an important role of niche preference in shaping AGF community, where hosts appear to select for specific AGF genera. Further, our ability to isolate and characterize T-AGF provided possible clues to the specific physiological and metabolic properties underpinning the niche-

based preference pattern observed. Specifically, optimal growth temperature of 30°C (as opposed to 39°C for mammalian AGF) suits the poikilothermic tortoise with lower gut temperature, and the slower growth rate could be useful for long-term colonization in tortoises, given the typically long food retention times observed in their GIT tracts (8-18 days in tortoises, compared to ~60 and ~40 hours in cows and horses, PMID: 33801700). As well, the observed limited CAZyme capacity in T-AGF could render them at a disadvantage in the highly competitive, prokaryotes-rich GIT tract of mammals, with relatively fast food retention times.

On the other hand, it appears that in a few samples (2/11), the reviewer's view regarding the functional interchangeability of AGF is certainly justified. The prevalence of AGF communities similar to mammalian AGF communities in few tortoises could be a reflection of a neutral selection process, where community structures are independent of species traits and governed mostly by stochastic processes. As such, it appears that both deterministic and stochastic processes play a role in shaping AGF-communities to different extents in various tortoise samples analyzed.

We took several steps to address the reviewer's comments in the revised manuscript:

1. We removed several statements that could be interpreted as advocating for a strict host-fungal evolution. These include: "*It is currently unclear whether T-AGF are the direct ancestors providing the seed for M-AGF, or whether M-AGF evolved from other yet-undiscovered extinct or extant ancestors*"- L522-524 in the original manuscript", and "the Order *Testudines* (with 13 other families encompassing side-necked turtles, softshell turtles, sea turtles, and others) is much older, evolving in the Late Triassic (237-201 Mya); and some of its extinct members (e.g. *Proganochelys*) were known to be herbivores, with a digestive process highly similar to extant tortoises"- L65-68 in the original manuscript).
2. We further provided an in-depth discussion that articulates our arguments (L535-563 in the revised manuscript), clarifying the possible role of selection versus cross-jumping and repeated secondary colonization in shaping AGF communities in tortoises. Specifically, we state that the results indicate that both stochastic and deterministic processes play a role in shaping AGF communities to varying extents in tortoises. In most samples (nine out of eleven) niche-based selection predominates, resulting in a community dominated by T-AGF), whereas the prevalence of AGF community similar to mammalian AGF communities in two out of eleven tortoise samples is a reflection of a neutral selection process, where community structures are independent of species traits and governed mostly by stochastic processes.

The results are consistent with recent and repeated colonization of tortoise guts by anaerobic fungi that are common in mammals and vice versa rather than with maintenance of genera of anaerobic fungi starting with early amniote evolution and continuing to modern tortoises. Comparing Figs. 1A and 1B, it looks like fungus NY54 is most common in tortoises and it also occurs widely across mammals. NY56 and NY36 predominated in one tortoise each, and appeared sporadically in mammals. And two tortoises had what appear to be a population of typical mammalian anaerobic fungi.

Thank you. As we describe above, the reviewer's thoughtful comments led us to rethink the ecological and evolutionary implications of our findings.

2. Line 65. Tortoises are herbivores with digestive systems that can accommodate anaerobic fungi, but to argue that the anaerobic fungi have been in turtle digestive tracts

for longer than the age of the Testudinidae, it would be important to present evidence that ancestral turtles were herbivores. Even if SOME extinct turtles were herbivores, it would be important for the authors to provide evidence against the possibility that ancestral turtle lineages should be reconstructed as aquatic carnivores. Within turtles, the tortoise lineage is only about 40 Myr old. The recent relatively origin of the tortoises suggests that their pool of anaerobic fungi are not ancestral, but the result of secondary colonization.

Thank you, as stated above, we agree with this assessment. We can see how the statements in L65-68 in the original manuscript could imply that “anaerobic fungi have been in turtle digestive tracts for longer than the age of the Testudinidae”. These sentences were removed in the revised manuscript.

3. The absence of evidence of herbivory as the ancestral state through all of the ancestors connecting tortoises with the first turtles is a problem for this argument:

Phylogenomic and molecular clock timing analysis estimated an evolutionary time of 112 Mya, and 104 Mya for the genera *Astrotestudinimycetes*, and *Testudinimycetes*, respectively. Such estimates predate the evolution of all current mammalian families known to harbor AGF (59-65). More importantly, it coincides with an increased diversification in fossil records of extinct turtle lineages during the lower Cretaceous (145-100 Mya).

Thank you. These sentences, part of the opening statement in the discussion in the original manuscript, have been removed from the revised manuscript. As explained above, this section has been rewritten to put more emphasis on explaining the community structure patterns observed in tortoises and the possible interpretation of such patterns from an evolutionary and ecological point of view. (L535-563 in the revised manuscript).

4. The study does not control for sampling depth, and the substantial editing needed to take this into account would result in a much different manuscript. Fungi from 11 tortoises in a zoo (this study) are being compared with fungi from many studies from varied mammals. This could explain low diversity in poorly sampled tortoises vs high diversity well sampled mammals.

Thank you. The reviewer contends that the comparative alpha diversity results (Figure 2A) could be influenced by disparities in sample depth/size, and origin.

Regarding sample size, we assume the reviewer refers to the number of analyzed samples. While it is true that hundreds of mammalian samples were examined in prior studies, a subsampling strategy was used to control for disparities in the number of samples in our analysis. Figure 2 compares 11 tortoises samples to datasets of relatively comparable size, comprising 24 deer, and 25 each of cattle, goats, and horses. This is clearly stated in L131-135 in the revised manuscript.

We would like to clarify that alpha diversity measures reported here (Sobs, Shannon, Simpson, and Inv. Simpson) were individually computed for each sample, e.g., a Shannon index value was computed for each of the 11 tortoise samples, a separate Shannon value for each of the 25 cattle samples and so on. These individual values were then plotted in the box and whisker plot of Figure 2a. Therefore, while an uneven number of samples could be present per animal (11 for tortoises, compared to 24-25 in other animals), and while additional tortoise samples would certainly have provided a better view of the alpha diversity of AGF in tortoises; the individual values of these diversity indices are not directly affected by the number of samples analyzed. We stress that we did not pool sequences from different samples to estimate the collective pan

diversity of AGF in an animal species. Rather we are reporting alpha diversity values per individual sample.

We would also like to clarify that the reference mammalian samples did not originate from “many studies from varied mammals” as the reviewer states. All samples originated from a single global survey study (PMID: 37365172), that was conducted in our laboratory. Both studies have followed the same sampling, amplification, sequencing, and sequence analysis protocols, thus minimizing interlaboratory differences and making the results more comparable. We added wording to highlight that in the revised manuscript (L129-131 in the revised manuscript).

Another issue the reviewer might be referring to/concerned about is the fact that all tortoise samples originated from zoo animals, compared to global mammalian samples, many domesticated in farms and many wild. This argument is certainly well-taken. In the revised manuscript, we acknowledge the difficulty of sampling wild tortoises, as many are endangered, and state that more work is needed (L564-573 in the discussion). Further, we conducted additional analysis, where we compared alpha and beta diversity measures in a subset of mammalian samples we generated in (PMID: 37365172) that were obtained from the Oklahoma City Zoo, the same zoo where 10/11 tortoise samples were obtained. The analysis again showed that tortoise AGF alpha diversity is significantly lower than the selected Bovidae AGF alpha diversity, even when samples originated from the same zoo. These additional results are now presented in L136-138 (methods), L385-391, and Figure S2 (Results).

5. Due to the sampling problem, the following are unsupported:

Assessment of alpha diversity patterns indicated that tortoises harbored a significantly less diverse AGF community when compared to placental mammals (p-value <0.04) (Figure 2a). Fig. 1B and 2A show low diversity of anaerobic fungi in 8 tortoise fecal samples but alpha diversity in two of the tortoises was as high or higher than for mammals (Fig. 2a) and in a third, it was about as high as in many mammals.

Thank you. We hope our argument for point 4 above provided better context. We agree with the reviewer that the sentence as written is inaccurate. We rewrote this sentence in the revised manuscript. It now reads: “Lower AGF diversity was observed in 8 out of 11 tortoise fecal samples, but alpha diversity in three samples (including the two samples that showed a community dominated by AGF genera typically encountered in mammals) was as high or higher than mammals (Fig. 2a)”. L377-380 in the revised manuscript.

6. It is not reasonable to attribute the difference to host class when sampling of mammals and tortoises was completely different:

Host class (Mammalia versus Reptilia) explained 47.6% of the variance (adonis p-value = 0.001).

Thank you, we removed this sentence in the revised manuscript and only stated that PCoA showed a distinction between tortoise and mammalian AGF mycobiomes (Figure 2b-c).

7. The collective predominance and ubiquity of these genera is in stark contrast to their rarity and low relative abundance, when encountered, in mammalian hosts (Figure 1b, (27)).

Thank you, this sentence (and the entire paragraph it was part of) was removed in the revised manuscript.

8. The issue of possible contamination must be addressed. No data are provided to document absence of contamination. Using only transcripts, distinguishing between genes that are from contaminating bacteria embedded in the fungi or closely adhering to the fungi vs genes that are horizontally transferred from bacteria to become part of the tortoise genome is difficult.

Thank you, we agree with the reviewer that contamination is an extremely important issue that needs to be addressed in any HGT study. In prior studies on the occurrence of HGT in AGF conducted by our research group (PMID: 31126947), we have addressed this issue in great detail. We should have provided more information on how this was conducted in the current manuscript, rather than simply referencing this paper. For that, we apologize. In the revised manuscript, we added details to the methods, highlighting all safeguards taken to ensure that the frequency and incidence of HGT events reported here are not due to bacterial contamination of AGF transcripts. (L254-267 in the revised manuscript)

In addition to methodological issues, we contend that the pattern of HGT observed in this study provides a strong argument that the results obtained reflect a true HGT process, and not due to contamination. First, all HGT events observed here (both the identity of genes and the identity of donors for each event) are similar to those observed in prior AGF-HGT studies conducted by our group and other groups (PMID: 31126947 and 28555641). This provides a strong indication that the results are a true reflection of HGT, rather than random contamination. Such a pattern could not certainly be obtained via random bacterial contamination where various donors and various genes would be observed due to contamination in different studies. Second, while genomes of these specific T-AGF are not yet available, analysis of HGT events in genomes of other AGF have clearly demonstrated that HGT genes identified are located on contigs in which they are flanked with genes of eukaryotic origin (PMID: 31126947), further demonstrating that they belong to AGF, and are not contaminants.

In the results section, we added wording to highlight that to the original statement that “within the limited number of HGT events identified in T-AGF, the majority (30/35) were also identified in M-AGF (44); and virtually all of which (29/30) had the same HGT donor (Table 1)”, we added the statement that “This consistency in HGT events (i.e. same genes, same donors) between both T-AGF genera as well as between T-AGF and M-AGF taxa examined argues against potential bacterial or archaeal contamination as a source of such transcripts” L467-470 in the revised manuscript.

The reviewer raises the possibility that contamination could be due to tight adherence to surface or even endophytic bacteria. This is an interesting and very astute observation. To our knowledge, tight adherence of bacteria to the surface of AGF, or the occurrence of endophytic bacteria have never been shown in any AGF strain isolated so far, and genome sequences consistently lack evidence for such occurrence (PMID: 28555641 and 23709508). Assuming these bacteria do exist, polyA tail selection during mRNA preparation and continuous culturing in triple antibiotic media (PMID: 27288952) should have inhibited their growth. In the unlikely scenario that these potential contaminants survived antibiotic treatment and their mRNA was partially retained post polyA tail selection, we posit that bacterial genes encountered and incorrectly identified as horizontally transferred would represent a random and disparate collection of genes from one or few microorganisms. This is clearly not the pattern seen in this study or prior AGF-HGT studies, in which specific genes (many of which have logical reasons to

be attained in AGF, as we describe in the results and discussion sections) from a wide range of bacteria (and eukaryotes) are detected in our analysis.

9. More information is needed about the quality of mRNA libraries from the tortoises. Authors mention that BUSCO transcripts were used to assess mRNA completeness, but (unless I missed them) they do not present the results. BUSCO are conserved and presumably highly transcribed genes that may not provide a good sense of diversity of CAZy mRNA recovered.

Thank you. In the revised manuscript, BUSCO results are presented in a new table (Table S1). While it is impossible to assess whether we missed something in a specific library, the highly similar CAZyome patterns (Figure 4, Dataset 4) between the five strains of *Testudinimyces* and the two replicates of the *Astrotetudinimyces* suggest that the results presented accurately reflect the CAZyome of these genera.

10. Statistics, language and analysis

• Line 348: "Community assessment using PCoA constructed using phylogenetic similarity-based weighted Unifrac..." wording is confusing.

We added parentheses "Community assessment using PCoA (constructed using phylogenetic similarity-based weighted Unifrac) confirmed the clear distinction....." to break down the sentence and increase clarity (L378-380 in the revised manuscript).

• Line 352: Does the DPCoA show that NY36, NY54, NY56 and the paucity of all other AGF is responsible for community structure distinction?

Yes, DPCoA uses both abundance and phylogenetic information about the samples, allowing both the samples and the taxa to be plotted on the same coordinate space, and thus the Euclidean distance between samples or their group centroids and AGF genera could be compared. Thus, AGF genera with Euclidean distances close to group centroids or samples are considered to contribute more to the community structure of the group.

• Line 599: "Results of two-tailed ANOVA..." incomplete sentence in figure legend. Should include that significance is shown above the boxplots, and the test that was used (t-tests? ANOVA isn't used for pairwise comparisons), as in Fig 1E. Also include what asterisks mean.

Thank you, yes, the sentence was incomplete. We added "are shown above the boxplots" in the revised manuscript (L672-675). The test used is Wilcoxon signed rank test for pairwise comparison. This has been corrected in the revised manuscript. Also, the meaning of asterisks was added (* denotes p values of $0.01 < p < 0.05$ and ** denotes $0.001 < p < 0.01$ (L675 in the revised manuscript)

• Line 603: "precentage" typo.

Sorry, corrected.

• In Figure 2A: Multiple pairwise t-tests increases chances of type I errors. Consider using post hoc analysis (e.g. Tukey's test).

Thank you. We agree that Post Hoc Tukey is needed for multiple comparisons; however, here we are conducting comparisons of tortoise sample alpha diversity to those in cattle, sheep, goats, or

horses. In this case, we believe that the Wilcoxon signed rank test is sufficient, and the p-values shown are for tortoise-cattle, tortoise-goat, tortoise-sheep, tortoise-horse only, and no other pairwise samples were compared here.

• Consider including what steps were taken to guard against detection of false HGT events due to bacterial contamination.

Thank you, as stated above, we have included this information in the revised manuscript (L252-265).

• Fig 2D vs Fig 4B: why is 2D a DPCoA whereas 4B is a PCoA?

Fig.4b is a Principal coordinate analysis plot that was generated using the abundance of different GH families in the transcriptomic datasets studied. The ordination is plotted as a biplot to show the position of the AGF (color-coded by genus) as well as the GH families responsible for that ordination. In Figure 2, the plot is, in fact, a DPCoA (double principal coordinate analysis) constructed with weighted Unifrac distances between samples. DPCoA needs taxonomic information (in the form of a phylogenetic tree that shows the relationship between genera present in the communities compared). It is not possible to use DPCoA for Figure 4b because that would require taxonomic information between the GH families.

Other points.

Use appropriate cladistic language to avoid incorrect interpretations of evolutionary history.

11. *Khoyollomyces* is no earlier evolving than any of the other anaerobic fungi because all originated from a common ancestral species that existed at a point in geological time. It would be correct to say that the most ancient divergence in Neocallimastigomycetes was between *Khoyollomyces* and all other known anaerobic taxa. genus *Khoyollomyces* recently identified as the earliest evolving AGF genus (55).

Thank you, correct. We changed it to “It is interesting to note that the most ancient divergence in Neocallimastigomycota was between *Khoyollomyces* and all other known anaerobic taxa (55)” (L364-366 in the revised manuscript)

12. When giving an age of a group, it is essential to specify which event in the group's origin is being dated. So was the earliest phylogenetic divergence among the aerobic fungi of tortoises 112.19 Ma, or was the divergence of the lineage of tortoise anaerobic fungi from the lineage of mammalian anaerobic fungi 112.19 Ma?

The former: “The earliest anaerobic divergence among the anaerobic fungi of tortoises was 112.19 Ma”. (L406 in the revised manuscript).

13. Fig. 1A. Scale lacks units. Should be in millions of years.

As suggested by reviewer #3, the tree was downloaded from timetree.org and the scale is better off removed from the figure. We have removed the scale from the revised manuscript.

14. Fig. 1a. Would better match order of presentation if legend were flipped to show % abundance on left, AGF genera on right.

Thanks, we agree. Changed in the revised manuscript.

15. Fig. 1b, right panel. Are the y-axis scales correct?

Yes, we apologize, the scales have been incorrectly labeled in two out of the three graphs. This is corrected in the revised manuscript. Thank you

Reviewer #2 (Remarks to the Author):

Summary: Anaerobic gut fungi (AGF) are a unique group of fungi that help breakdown indigestible plant fibers into nourishing food for their mammalian hosts, e.g., ruminants and hind-gut fermenters. A recent study by Meili and colleagues in 2023, which is also cited as Ref. 27, found that the AGF in mammals (M-AGF) started to diversify about 67 million years ago. Here, the same authors used metagenomic sequencing, culturing, and transcriptome profiling to discover new AGF species in the gut of tortoises. They first measured the AGF community composition and abundance from tortoise feces using rRNA sequencing (Figures 1-2). Excitingly, their analysis revealed that these new AGF species in tortoises, known as T-AGF (NY54, NY56, NY36), are an early branch of AGF that is distinct from the mammalian AGF (M-AGF). This discovery has the potential to provide fresh insights into how AGF and their animal hosts have evolved together.

The authors then isolated and cultured NY54 and NY36 from tortoise feces, and measured the expressed gene content in these T-AGF using RNA-seq. These data were used to find and align 88 conserved genes across M-AGF, T-AGF, and distant chytrid fungi to build a time-calibrated phylogenetic tree (Figure 3). This robust phylogeny confirms that T-AGF are an early-diverging lineage that diverged about 104 MYA from the common AGF ancestor shared with M-AGF. The authors then used this transcriptome dataset to show there are fewer T-AGF enzymes associated with cellulose and hemicellulose metabolism relative to M-AGF (Figures 4-5). Thus, T-AGF appear to be less capable of plant carbohydrate degradation compared to M-AGFs due to decreased CAZyme presence and reduced growth on specific plant carbohydrate media. Last, the authors identify 35 potential genes that arrived via horizontal gene transfer (HGT) into T-AGF to impart new functions (Table 1). Most of these HGT are shared with M-AGF suggesting that they were present in the ancestor of both lineages, and many of their functions are associated with carbohydrate catabolism, fermentation, and anaerobic metabolism.

This is an important manuscript because it provides solid evidence that the symbiotic association of AGF with herbivorous animals is more ancestral than previously thought. This paper will be of keen interest to general readers such as microbiologists, evolutionary biologists, and researchers interested in fungal-animal symbiosis. This manuscript will catalyze future research to uncover AGF in other herbivores and elucidate the role of early-diverging AGF in plant digestion within non-mammalian species.

We thank Reviewer 2 for the detailed and accurate summary!

The AGF from Ploughshare and Galapagos Zoo_604 were strikingly different from other tortoise samples and closer to M-AGF (Anaeromyces, Neocallimastix, Orpinomyces, etc.). This raises questions about whether these AGF were obtained from tortoise feces without

any external contamination. It also makes us wonder if these particular tortoises were sick or might have had unusual digestive abilities when it comes to breaking down plant carbohydrates. We were hoping for a more in-depth discussion regarding this distinct bimodal difference in the composition of T-AGF communities.

Thank you. Yes, the AGF community structure was indeed strikingly different in these two samples, compared to all other Tortoise samples. We acknowledge that such an issue was not adequately discussed in the original version of the manuscript, something that the two other reviewers have also noticed and commented upon.

The reviewer provides some possible interpretations of this observation. The first is external contamination during sampling. We believe that the sampling process was not associated with any contamination. We have been working closely with the highly skilled personnel in the Oklahoma City Zoo for over a decade (see PMID: 37365172 and 20410935). We are confident that the zoo personnel strictly adhere to the sampling protocol provided, where freshly deposited feces of a single animal are promptly deposited into a single sterile container, with no inclusion of traces of other samples or even soil/dirt particles. As well, all tortoises are housed in a single building (The Herpetarium) in the Oklahoma City Zoo that is separated from all other mammalian habitats and enclosures. All samples were also inspected and appeared to be a single sample judging by their shape and consistency. As such, we are confident that it is not a case of sampling error/external cross-fecal contamination. In the revised manuscript, we added these clarifications to the revised manuscript (L83-87 in the revised manuscript).

Laboratory contamination during PCR amplification would be an additional possibility. However, again, we took all necessary steps to ensure that this is not the case. Negative (no DNA) controls were conducted. In the revised manuscript, we added, “Negative (no DNA) controls were conducted with all amplifications (L111-112 in the revised manuscript).

The reviewer suggests sickness or possession of special digestive abilities by these tortoises as a possible explanation. Upon checking with the OKC Zoo personnel, all sampled tortoises did not suffer from any discernible sickness at the time of sampling. We added a sentence to state that in the revised manuscript (L87-88 in the revised manuscript).

Regarding possible differences in the digestive abilities of the tortoises: While detailed comparative assessments of GIT tract characteristics in tortoises are scarce, data (some summarized in Table S3) indicate that collectively, no drastic differences in the GIT tract architecture or food retention time exist between Ploughshare and Galapagos tortoises on one hand, and all others on the other. The only plausible speculation we can provide is that the more versatile species feeding preferences (i.e., the fact that ploughshare and Galapagos tortoises are capable of frugivory, graminivory, and folioivory) or their original island habitat could drive such differences. However, this is purely speculative, and in the case of the Galapagos tortoise, another individual sampled had a community dominated by a T-AGF (NY54).

While we cannot pinpoint the exact reason for these observed differences in community structure, we certainly agree that this is an issue worth further examination and should be discussed in more detail. As we describe above, we provide a more in-depth discussion of this issue in the opening paragraph of the discussion in the revised manuscript (L557-563 in the revised manuscript).

The authors might consider including data showing that T-AGF are unable to grow efficiently on plant carbohydrates as M-AGF in Figure 4 because this would complement and reinforce the CAZyme analysis.

Thank you very much. The slower cellulose-degradation ability and the lack of xylan degradation abilities in T-AGF taxa, compared to mammalian AGF-taxa has been tested, and the results are stated in L384-387 in the original manuscript (L443-448 in the revised manuscript), as well as in Figure S6. Additional characterization efforts are also available in the recent taxonomy manuscript naming these new isolates as two novel genera (PMID: 37252853), which is also referenced in the same paragraph

Table 1 shows that the ancestor of T-AGF and M-AGF acquired new genes through horizontal transfer that likely played a role in carbohydrate catabolism, fermentation, and adaptation to an anaerobic gut environment. Do T-AGF have intact mitochondria, degraded mitochondria, or hydrogenosomes? This type of data could provide important evidence on the timeline of AGF adaptations to the herbivore gut.

Thank you so much for the excellent suggestion. M-AGF possess double-membrane hydrogenosomes, an organelle whose main functions are substrate-level phosphorylation and H₂ production.

In the revised manuscript, we undertook additional analysis to examine the occurrence and nature of T-AGF hydrogenosomes, as well as to compare their contents to M-AGF hydrogenosomes. Comparison of the predicted hydrogenosomal content of one representative of each T-AGF genus, to an M-AGF strain showed similar patterns between M-AGF and T-AGF, with a near complete hydrogenosomal protein import system, chaperones/co-chaperones, mitochondrial peptidases, and mitochondrial transporters (new Dataset 3). Hydrogenosomes in M-AGF are the sites of multiple metabolic processes, including pyruvate metabolism, ATP production via substrate-level phosphorylation, regeneration of reduced electron carriers, some amino acid biosynthesis, fatty acid biosynthesis, and Fe-S cluster assembly. Our comparative analysis showed a similar pattern in T-AGF predicted hydrogenosomes, with all these functions predicted to be hydrogenosomal. While unusual, the only observed difference was the possession of T-AGF (but not M-AGF) of several hydrogenosomal peptides assigned to DNA repair and recombination, several translation factors and proteins, and several mRNA and tRNA biogenesis proteins hinting at a less degraded organelle. The new information is presented in L283-291 (methods) and L415-436 (results), and the list of identified hydrogenosomal proteins is presented in an additional dataset (Dataset 3).

Minor comments:

Figure 2C. The color for tortoise data points is hard to see at the zone of overlap with mammalian data points. Can tortoise points be highlighted in a way to distinguish them from the other data? E.g., dark circle around tortoise points, bring them to the foreground, no outline for mammalian points.

Yes, thank you for the suggestion. We have surrounded the tortoise sample points with a dashed stroke in Figures 2C and 2D in the revised manuscript.

Line 165: Figure 3 contains 8 T-AGF species, including T. gracilis sp. K01. However, the sentence says 7 species and omits K01? The Materials & Methods continue to say 7 transcriptomic datasets. Is Figure 3 wrong, or is Materials & Methods wrong?

Thank you for the comment. We apologize for the confusion. The M+M was incorrect. Eight species was used for phylogenomic analysis. We changed the number from 7 to 8 (L170 and L186 in the revised manuscript). It is worth noting that K01 species was not included in

subsequent analysis, due to uncertainty of its exact origin (which sample it came from).

Line 186-187: The authors state that the "final alignment file included 88 genes ... and comprising more than 150 nucleotide sites." Did the authors mean to say 150 nucleotide sites per gene?

We apologize for the confusion. The total number of included nucleotide sites was 37,044. This has been corrected in the revised manuscript (L190-192 in the revised manuscript).

Line 360: Missing "c" in -myces of Astrotestudinimycetes.

Corrected. We apologize.

Line 580: There should be a period after timetree.org before the next sentence.

Thank you. Added.

Reviewer #3 (Remarks to the Author):

The manuscript describes the anaerobic gut fungi (AGF) of tortoises and compares them to those found in mammalian hosts. The work uses amplicon sequencing, transcriptomics, proteomics, and phylogenomics to analyze fecal samples from several tortoises and concludes that the predominant AGF in tortoises are distinct from those predominant in mammals, and likely diverged some time before the massive evolutionary diversification of mammalian herbivores following the late Cretaceous extinctions. It is also concluded that tortoise AGF have fewer CAZymes and less horizontal gene transfer (HGT) than seen in mammalian-host AGF, and some interesting speculations are made as to why.

The work expands what is known about AGF from mammalian to non-mammalian herbivores (tortoises). In doing so it provides insights about the evolution of animal-fungal symbioses as they relate to periods of extinction and/or animal species diversification. It also expands functional and ecological knowledge of Neocallimastigomycota, an interesting phylum of early-diverging fungi which are not well understood.

The manuscript is well written and the data and methods presented appear to provide enough detail to reproduce the work, and support the claims and conclusions sufficiently well.

Thank you very much for the positive assessment and recognizing the significance of our findings!

Minor comments:

- In Fig 1a, many of the tortoises have surprisingly low AGF diversity. Most of the tortoises are from the same zoo, so could the environment and factors such as shared pens or feedstocks be influencing AGF diversity compared to animals in the wild? Why are a few tortoise's AGFs extremely diverse in comparison? I understand from the manuscript that studying wild tortoises is difficult, but discussion of caveats would be helpful.

Thank you, yes this is related to reviewers 1 and 2 comments regarding the observed differences in diversity and community structure between samples dominated by T-AGF and samples dominated by M-AGF. Again, we acknowledge that this observation has not been adequately discussed in the original version of the manuscript. As we describe above, additional discussion

of the observed pattern was provided in the discussion section of the revised manuscript (L557-563 in the revised manuscript).

- Also in Fig 1a: please explain the branch lengths, or omit scale bar since it apparently was an externally downloaded tree. Also "composition" is misspelled. Fig 1a doesn't really show 'columns' as indicated in the legend, rather 'bars' since they are sideways.

Thank you. Yes, the tree was downloaded from timetree.org. We omitted the scale bar in the revised manuscript. We also fixed the typo and changed “columns” to “bars”.

- Table S1 would be easier to follow if the organism names matched the tree in Fig 1a were shown.

Thank you. Yes, we absolutely agree. Names in the supplementary table (now Table S3) have been corrected to match the names in the tree in Fig 1a in the revised manuscript.

- 387, 408, 412: Change "Such pattern..." to "Such a pattern" or "This pattern"

Thank you. Changed to “this pattern” in the first and third locations. In the second location, the wording was removed, and additional text was included to address reviewer 1 inquiries regarding the interpretation of the HGT patterns observed.

- 427-430: These statements need references

Thank you. References provided in the revised manuscript.

- 469: I don't see a Figure S7 in the supplementary. It only goes up to S6.

We apologize for the truncation of the supplementary file. The full file is now uploaded.

- 552-554: Needs references

Agreed, reference provided.

- 556-559: Needs references

Agreed, reference provided.

Again, we thank the reviewers for their comments and suggestions, which have greatly improved the quality and readability of the manuscript. We hope we have satisfactorily addressed their comments.

Sincerely yours,

Dr. Noha Youssef

REVIEWER COMMENTS

Reviewer #1 (Remarks to the Author):

The authors made strides in addressing reviewer comments, but the interpretations of the results are still not fully consistent with the data. The authors and I agree that their data show that anaerobic fungal genera are shared by tortoises and mammals. The only possible explanation for AGF presence both in tortoises and mammals is host jumping.

The text is still not clear about implications. If ancestral anaerobic fungi presumably also jumped to any available host that had an anaerobic gut and a diet rich in plant material. This is an appealing view of these highly specialized fungi because it means fungal populations could have survived through extinctions of individual host species.

The evidence that anaerobic fungi are not as abundant in tortoises as in mammals is convincing (Fig. 1e), but that tortoise anaerobic fungi are fundamentally different from mammalian fungi is not yet fully convincing.

1. The problem of unequal samples may have distorted the results especially when comparing diversity of enzymes. Isolates from mammals are diverse and extensive, while samples from tortoises are few and closely related to one another. The CAZY study compares transcriptomes of 52 non-tortoise isolates with 8 tortoise isolates. The tortoise isolates must all fall into one of two, closely related species, while the non-tortoise isolates are presumably diverse species. The study should use rarefaction to control for sample number, and it should control for phylogenetic depth of sampling. So if 8 isolates related to NY36 and 54 are available from tortoises, those isolates should be compared with randomly selected 8 isolates from two similarly distant species of other anaerobic fungi. It's necessary to rule out the possibility that sampling the same or highly similar tortoise fungi explains their relative lack of enzymatic diversity.

2. A supplemental likelihood tree without the time estimates should be provided because divergence order is an important point in the paper. Line 367: It is interesting to note that the most ancient divergence in Neocallimastigomycota was between *Khoyollomyces* and other known anaerobic taxa." In contrast to the note on line 367 and to the LSU tree in fig. 1, the dated tree shows the tortoise isolates as the basal anaerobic fungi. A multilocus ML tree without the dating constraint is needed to better resolve the divergence order and support or refute the idea that the earliest divergence was between the fungal clades that predominate in tortoises and the clades that predominate in most mammals.

An ML tree is needed because dating reduces by one the parameters used to estimate branch lengths. A dated phylogeny typically (always?) has a lower likelihood than the undated phylogeny, which may be fine as long as the branching order of the dated and undated ML trees match. But if rates and modes of evolution vary across clades in ways that are difficult to model, the extra parameter allowed in an undated phylogeny may result in a different and more plausible branching order.

3. More analysis is needed to support the interesting possibility (added in revision) that T130A.3 and B1.1 have mitochondrial genomes. Even reduced mitochondrial genomes include protein translation machinery, so it may be possible to find fragments of AT-rich mitochondrial ribosomal RNA genes among transcripts. If genes are mitochondrial and their proteins function in mitochondria, they won't have signal peptides. Mitochondrial GC content is usually low compared with nuclear encoded genes. It should be possible to use primers developed from putative mitochondrial genes to show that more than one of the genes is on the same contig. Phylogenies may show that the putative mitochondrial genes cluster with mitochondrial genes from aerobic relatives.

Specific, minor criticism:

4. Correct the period. 67 Mya is Late Cretaceous, not Paleogene (see <https://stratigraphy.org/ICSchart/ChronostratChart2023-04.pdf>): "from the early Paleogene (67

Mya) to the early Cretaceous (112 Mya)"

5. Correct the implication: 67 Mya predates the K-Pg extinction event, so the following is incorrect: "Prior efforts based on available M-AGF taxa estimated an AGF 588 divergence time of 67 Mya 27, 32. Such estimate post-dates the K-Pg extinction event"

6. Clarify. The meaning of 'independently evolved' is unclear in the quote below. Anaerobic ability did not evolve independently; it has to be assumed to have evolved once in this fungal group. The divergence may have happened early, but mammals are old enough so it could have originated among them.

"Our results describe two distinct, deep branching lineages that independently evolved 37-45 Mya prior to these events in a non-mammalian host.

7. The temperatures listed below are high or very high, so the text and the numbers are at odds: "...nature of the host, and the fact that the tortoise gut community is often exposed to lower and 165 variable temperatures, we enriched for tortoise-associated AGF at a range of temperatures (30oC, 39oC, and 42oC).

8. Missing end of sentence:

435 translation factors and proteins, and several mRNA and tRNA biogenesis proteins hinting 436 (Dataset 3).

Reviewer #2 (Remarks to the Author):

The authors did an excellent job of addressing my questions and revising the manuscript accordingly.

Reviewer #2 (Remarks on code availability):

There is no code associated with this manuscript.

Reviewer #3 (Remarks to the Author):

My concerns have been adequately addressed and the new manuscript is much improved.

January 23rd, 2024

Response to reviewers' comments on Nature Communications NCOMMS-23-42309A

Reviewer #1 (Remarks to the Author):

The authors made strides in addressing reviewer comments, but the interpretations of the results are still not fully consistent with the data. The authors and I agree that their data show that anaerobic fungal genera are shared by tortoises and mammals. The only possible explanation for AGF presence both in tortoises and mammals is host jumping. The text is still not clear about implications. If ancestral anaerobic fungi presumably also jumped to any available host that had an anaerobic gut and a diet rich in plant material. This is an appealing view of these highly specialized fungi because it means fungal populations could have survived through extinctions of individual host species.

Thank you very much for the positive reception of our prior revisions. We, of course, agree that host-jumping occurs by members of the Neocallimastigomycota across various hosts. We agree with the reviewer that the implications of such ability need to be spelled out. In the revised manuscript, we added the statement that “The implications of this cross-colonization and host-jumping ability are significant, since it would enable various fungal taxa to survive through extinctions of various herbivorous hosts.” L553-555 in the revised manuscript.

The evidence that anaerobic fungi are not as abundant in tortoises as in mammals is convincing (Fig. 1e), but that tortoise anaerobic fungi are fundamentally different from mammalian fungi is not yet fully convincing.

1. The problem of unequal samples may have distorted the results especially when comparing diversity of enzymes. Isolates from mammals are diverse and extensive, while samples from tortoises are few and closely related to one another. The CAZY study compares transcriptomes of 52 non-tortoise isolates with 8 tortoise isolates. The tortoise isolates must all fall into one of two, closely related species, while the non-tortoise isolates are presumably diverse species. The study should use rarefaction to control for sample number, and it should control for phylogenetic depth of sampling. So if 8 isolates related to NY36 and 54 are available from tortoises, those isolates should be compared with randomly selected 8 isolates from two similarly distant species of other anaerobic fungi. It's necessary to rule out the possibility that sampling the same or highly similar tortoise fungi explains their relative lack of enzymatic diversity.

Thank you, we undertook additional analysis to address this possibility of sample size distortion.

To this end, in addition to the Tortoise_{all} Vs Mammalian_{all} CAZyome comparison (Figure 4a), we undertook four additional pairwise comparisons between Tortoise CAZyomes (n=7) versus an equal number of mammalian transcriptomes. Each M-AGF transcriptome dataset used originated from two different genera within a single AGF mammalian family to control for the phylogenetic depth of sampling. Specifically, the following pairwise comparisons were conducted

1. T-AGF CAZyome (n=7, two genera), versus M-AGF CAZyomes from the genera *Capellomyces* and *Anaeromyces* (n=9), both of which members of the family *Anaeromycetacea*.

2. T-AGF CAZyome (n=7, two genera), versus M-AGF CAZyomes from the genera *Caecomyces* and *Cyllamyces* (n=6), both of which members of the family *Caecomycetacea*.
3. T-AGF CAZyome (n=7, two genera), versus M-AGF CAZyomes from the genera *Orpinomyces* and *Pecoramyces* (n=8), both of which members of the family *Neocallimastigaceae*.
4. Because the family *Neocallimastigaceae* is extensive, we added another dataset of M-AGF transcriptomes for comparison with the genera *Neocallimastix* and *Feramyces* (n=8).

The results of these pairwise comparisons are described in the text of the revised manuscript (L466-468 in the revised manuscript) and added as a new supp figure (Figure S8). As well, the statistical significances of these differences are added in a new supplementary Table S5.

2. A supplemental likelihood tree without the time estimates should be provided because divergence order is an important point in the paper. Line 367: It is interesting to note that the most ancient divergence in Neocallimastigomycota was between *Khoyollomyces* and other known anaerobic taxa." In contrast to the note on line 367 and to the LSU tree in fig. 1, the dated tree shows the tortoise isolates as the basal anaerobic fungi. A multilocus ML tree without the dating constraint is needed to better resolve the divergence order and support or refute the idea that the earliest divergence was between the fungal clades that predominate in tortoises and the clades that predominate in most mammals.

An ML tree is needed because dating reduces by one the parameters used to estimate branch lengths. A dated phylogeny typically (always?) has a lower likelihood than the undated phylogeny, which may be fine as long as the branching order of the dated and undated ML trees match. But if rates and modes of evolution vary across clades in ways that are difficult to model, the extra parameter allowed in an undated phylogeny may result in a different and more plausible branching order.

Thank you, we agree. In the revised manuscript, we added an additional multi-locus ML tree as supplemental material (Figure S4). The results of Multi-locus and dating trees places the fungal clades predominating in tortoises as basal to all mammalian clades including *Khoyollomyces*.

The statement “It is interesting to note that the most ancient divergence in Neocallimastigomycota was between *Khoyollomyces* and other known anaerobic taxa” is presented early in the results and was meant to describe the current state of knowledge prior to conducting any analysis in this paper. In retrospect, its placement and lack of context is confusing. We apologize. In revised manuscript, the issue of phylogenetic placement and divergence order issue is clearly tackled only in one place in the results section (L406-410, including new information and reference for the new ML tree figure S4), and also discussed in one place only in the discussion section (L598-613).

3. More analysis is needed to support the interesting possibility (added in revision) that T130A.3 and B1.1 have mitochondrial genomes. Even reduced mitochondrial genomes include protein translation machinery, so it may be possible to find fragments of AT-rich mitochondrial ribosomal RNA genes among transcripts. If genes are mitochondrial and their proteins function in mitochondria, they won't have signal peptides. Mitochondrial GC content is usually low compared with nuclear encoded genes. It should be possible to use

primers developed from putative mitochondrial genes to show that more than one of the genes is on the same contig. Phylogenies may show that the putative mitochondrial genes cluster with mitochondrial genes from aerobic relatives.

Thank you for the interesting suggestion. We agree that the possession of a larger number of hydrogenosomal peptides in T-AGF when compared to M-AGF; as well as the fact that many of these additional peptides are assigned to DNA repair and recombination, translation and transcription proteins, and several mRNA and tRNA biogenesis proteins, raises the possibility of actual existence of mitochondrial genomes in T-AGF. Unfortunately, we do not feel that the nature of the current dataset (transcriptomes, rather than genomes) allows us to decisively test this hypothesis. Clearly, we cannot look for rRNA genes (DNA) in the transcriptomic datasets. We have looked at rRNA (RNA) in both our assembled transcripts dataset, as well as the raw reads, and could not find any. However, this could be clearly a function of the fact that a ribodepletion step is conducted during the process of preparing mRNA for transcriptomic sequencing.

The reviewer also suggests using GC content as a way to differentiate between genes/transcripts encoded in mitochondrial versus nuclear genomes. However, we note that the GC content of the AGF transcripts, including these newly isolated T-AGF (Average $23.3 \pm 3.7\%$) are extremely low to start with. The AGF have some of the lowest GC content transcripts in the fungal kingdom, rendering the use of this approach for differentiation difficult, and any results obtained would be quite putative.

We are currently working hard towards obtaining genome assemblies from these T-AGF strains. The large number of long repeats in the intergenic regions in AGF genomes necessitates the use of long read sequencing technology (rather than Illumina) to enable assembly. Unfortunately, obtaining high MW DNA is exceptionally hard for the AGF. We are making significant progress towards this goal, and we certainly intend to examine such possibility in the future. Finally, we, very respectfully, note that while this issue is extremely exciting, is not the main focus of this manuscript.

In the revised manuscript, we added this extra information and analysis to address this point.

We stated our argument that we are only hinting at the possibility of presence of mitochondrial genomes (L436-445). We state the difficulty with addressing that with transcriptomes, and report our inability to find rRNA in our dataset and why that is the case.

Specific, minor criticism:

4. Correct the period. 67 Mya is Late Cretaceous, not Paleogene

(see <https://stratigraphy.org/ICSchart/ChronostratChart2023-04.pdf>): "from the early Paleogene (67 Mya) to the early Cretaceous (112 Mya)"

Thank you, corrected to "from the late Cretaceous (67 Mya) to the early Cretaceous (112 Mya)". L32-33 in the revised manuscript.

5. Correct the implication: 67 Mya predates the K-Pg extinction event, so the following is incorrect: "Prior efforts based on available M-AGF taxa estimated an AGF

588 divergence time of 67 Mya 27, 32. Such estimate post-dates the K-Pg extinction event"

Correct, it is one year prior, not post, the K-Pg extinction event. We have changed the sentence to "Such estimate is close to the K-Pg extinction event (66 Mya)" (L601 in the revised manuscript).

6. Clarify. The meaning of 'independently evolved' is unclear in the quote below. Anaerobic ability did not evolve independently; it has to be assumed to have evolved once in this fungal group. The divergence may have happened early, but mammals are old enough so it could have originated among them.

"Our results describe two distinct, deep branching lineages that independently evolved 37-45 Mya prior to these events in a non-mammalian host.

Thank you, we agree that the word “independently” is incorrect here. We have removed it from the revised manuscript. As well, we agree with the reviewer that we cannot ascertain that the evolution of these new T-AGF genera occurred in a non-mammalian host (it could have occurred in any other host, including mammals as the reviewer state). We removed “in a non-mammalian host” from the revised manuscript (L604-607 in the revised manuscript).

7. The temperatures listed below are high or very high, so the text and the numbers are at odds:

"...nature of the host, and the fact that the tortoise gut community is often exposed to lower and

165 variable temperatures, we enriched for tortoise-associated AGF at a range of temperatures (30oC, 39oC, and 42oC).

Thank you, the temperatures listed are the temperatures at which we attempted to enrich for AGF from tortoises fecal samples. The lower two temperatures (30° C and 39° C) yielded isolates, while enrichments at 42° C did not. When the work commenced, we were not sure whether AGF, if encountered in tortoises, would be more adapted to the lower or higher temperature bounds within the broad variable internal temperature encountered by tortoises. Admittedly, 42° C is dangerously close to the lethal internal temperatures of most tortoises (around 43° C). We are just reporting what we have attempted.

To clarify how these enrichments at various temperatures fared, we added the sentences underlined below the following section in the results section:

“*Testudinomyces* and *Astrotestudinomyces* isolates were enriched at, and displayed optimum growth temperatures, of 30° C, and 39° C, respectively. On the other hand, despite repeated successful enrichment and isolation rounds (conducted at 39° C), isolates belonging to candidate genus NY56 have been extremely hard to maintain as viable cultures for subsequent analysis.

The names *Testudinomyces* and *Astrotestudinomyces* will henceforth be used to describe cultured strains belonging to NY54 and NY36 in the manuscript. Enrichments at 42° C did not yield any visible fungal growth.” (L398-404 in the revised manuscript)

8. Missing end of sentence:

translation factors and proteins, and several mRNA and tRNA biogenesis proteins hinting (Dataset 3).

Thank you, we apologize. The sentence has been amended to state “ hinting at a possible presence of an organelle genome (Dataset 3)” (L438-439 in the revised manuscript). Further, as stated in our response to point #3 above, this section has been expanded to add “Further analysis is needed to confirm this possibility (for example identification of mitochondrial rRNA and tRNA genes, or phylogenetic congruency of mitochondrial T-AGF genes with their aerobic counterparts). However, the nature of the current dataset (transcriptomic rather than genomic) and the fact that ribodepletion was applied prior to poly-A transcripts selection for mRNA

enrichment prior to RNA-seq would prohibit such analysis currently. Future availability of genomes from T-AGF isolates should allow further analysis of this interesting possibility.”.

Reviewer #2 (Remarks to the Author):

The authors did an excellent job of addressing my questions and revising the manuscript accordingly.

Thank you!

Reviewer #2 (Remarks on code availability):

There is no code associated with this manuscript.

Thanks, there is code associated with this and other manuscripts. The code availability is stated in the manuscript (L344-346).

Reviewer #3 (Remarks to the Author):

My concerns have been adequately addressed and the new manuscript is much improved.

Thank you!

Once again, we thank the reviewers for the time and effort spent on the manuscript. We hope these changes render the manuscript suitable for publication in Nature Communications.

Sincerely yours,

Dr. Noha H. Youssef
Professor of Microbiology
Oklahoma State University.

REVIEWERS' COMMENTS

Reviewer #1 (Remarks to the Author):

I agree that the authors have addressed my concerns and look forward to seeing the manuscript in print.